# Large-eddy simulation of radiation fog with comprehensive two-moment bulk microphysics: impact of different aerosol activation and condensation parameterizations

Johannes Schwenkel[1] and Björn Maronga[1,2]

[1]Institute of Meteorology and Climatology, Leibniz University Hannover, Hannover, Germany
[2]Geophysical Institute, University of Bergen, Bergen, Norway

*Correspondence to:* Johannes Schwenkel (schwenkel@muk.uni-hannover.de)

**Abstract.** In this paper we study the influence of the cloud microphysical parameterization, namely the effect of different methods for calculating the supersaturation and aerosol activation, on the structure and life cycle of radiation fog in large-eddy simulations. For this purpose we investigate a well-documented deep fog case as observed at Cabauw (Netherlands) using high-resolution large-eddy simulations with comprehensive bulk cloud microphysics scheme. By comparing saturation adjustment with a diagnostic and a prognostic method for calculating supersaturation (while neglecting the activation process) we find that, even though assumptions for saturation adjustment are violated, the expected overestimation of the liquid water mixing ratio is negligible. By additionally considering activation, however, our results indicate that saturation adjustment, due to approximating the underlying supersaturation, leads to a higher droplet concentration and hence significantly higher liquid water content in the fog layer, while diagnostic and prognostic methods yield comparable results. Furthermore, the effect of different droplet number concentrations is investigated, induced by using different common activation schemes. We find, in line with previous studies, a positive feedback between the droplet number concentration (as a consequence of the applied activation schemes) and strength of the fog layer (defined by its vertical extent and amount of liquid water). Furthermore, we perform an explicit analysis of the budgets of condensation, evaporation, sedimentation and advection in order to assess the height-dependent contribution of the individual processes on the development phases.

## 1 Introduction

The prediction of fog is an important part of the estimation of hazards and efficiency in traffic and economy (Bergot, 2013). The annual damage caused by fog events is estimated to be the same as the amount caused by winter storms (Gultepe et al., 2009). Despite improvements in numerical weather prediction (NWP) models, the quality of fog forecasts is still unsatisfactory. The explanation for this is obvious: fog is a meteorological phenomenon influenced by a multitude of complex physical processes. Namely, these processes are radiation, turbulent mixing, atmosphere-surface interactions, and cloud microphysics (hereafter referred to as microphysics), and which interact on different scales (e.g. Gultepe et al., 2007; Haeffelin et al., 2010). The key issue for improving fog prediction in NWP models is to resolve the relevant processes and scales explicitly, or - if that is not possible - to parameterize them in an appropriate way.

In recent years, various studies focused on the influence of microphysics on fog. In particular, the activation of aerosols (hereafter simply referred to as activation), which determines how many aerosols at a certain supersaturation get activated and hence can grow into cloud drops, is a key process and thus of special interest (e.g. Bott, 1991; Hammer et al., 2014; Boutle et al., 2018).

Stolaki et al. (2015) investigated and compared the influence of aerosols on the life cycle of a radiation fog event while using the one-dimensional (1D) mode of the MESO-NH model with a two-moment warm microphysics scheme after Geoffroy et al. (2008) and Khairoutdinov and Kogan (2000), and included an activation parameterization after Cohard et al. (1998). In other fog studies, using single-column models, different activation schemes such as the simple Twomey-power law activation in Bott and Trautmann (2002) and the scheme of Abdul-Razzak and Ghan (2000) (see Zhang et al., 2014) were applied. Furthermore, also more advanced methods such as sectional models have been used for an appropriate activation representation. Maalick et al. (2016) used the Sectional Aerosol module for Large Scale Applications (SALSA) (Kokkola et al., 2008) in two-dimensional (2D) studies for a size-resolved activation. Mazoyer et al. (2017) conducted, similar to Stolaki et al. (2015), simulations for the ParisFog Experiment with the MESO-NH (for more information to the MESO-NH model, see Lac et al., 2018) model, but using the three-dimensionsl (3D) Large-Eddy Simulation (LES) mode, and focusing on the drag effect of vegetation on droplet deposition. For the fog microphysics they used the activation parameterizations after Cohard et al. (2000) in connection with saturation adjustment. As outlines above, several different activation parameterizations have been employed for simulating radiation fog. This raises the question how different methods affect the structure and life cycle of radiation fog. Furthermore, schemes that parameterize activation based on updrafts (typically done in NWP models) might fail for fog. Such schemes derive supersaturation as a function of vertical velocity, which is valid for convective clouds that are forced by surface heating, but not for radiation fog, which is mainly driven by longwave radiative cooling in its development and mature phase (Maronga and Bosveld, 2017; Boutle et al., 2018).

Although great progress has been made to understand different microphysical processes in radiation fog based on numerical experiments, turbulence as a key process has been either fully parameterized (single-column models) or oversimplified (2D LES). Since turbulence is a fundamentally 3D process, the full complexity of all relevant mechanisms can only be reproduced with 3D LESs (Nakanishi, 2000).

Moreover, a disadvantage of most former studies is the use of saturation adjustment, which implies that supersaturations are immediately removed within one time step. This approach is only valid when the time scale for diffusion of water vapour (in the order of 2-5 s) is much smaller than the model time step. This is the case in large scale models where time steps are on the order of 1 min, but in LES of radiation fog, time steps easily go down to split seconds so that the assumption made for saturation adjustment is violated and might lead to excessive condensation (e.g. Lebo et al., 2012; Thouron et al., 2012). As a follow-up to these studies, who investigated the influence of different supersaturation calculations for deep convective cloud and stratocumulus, the present work investigates the effect of saturation adjustment on radiation fog.

As Mazoyer et al. (2017) and Boutle et al. (2018) stated that both LES and NWP models tend to overestimate the liquid water content and the droplet number concentration for radiation fog, the following questions are derived from these shortcomings:

(i) Is saturation adjustment appropriate as it crucially violates the assumption of equilibrium? How large is the effect of different methods to calculate supersaturation on diffusional growth of fog droplets?

(ii) As the number of activated fog droplets is essentially determined by the supersaturation, how large is the effect of different supersaturation modeling approaches on aerosol activation and therewith on the strength and life cycle of radiation fog (cf. Thouron et al., 2012)?

(iii) What is the impact of different activation schemes on the fog life cycle for a given aerosol environment?

In the present paper we will address the above research questions by employing idealized high-resolution LESs with atmospheric conditions based on an observed typical deep fog event with continental aerosol conditions at Cabauw (Netherlands).

The paper is organized as follows: section 2 outlines the methods used, that is the LES modeling framework and the microphysics parameterizations used. Section 3 provides an overview of the simulated cases and model setup, while results are presented in section 4. Conclusions are given in section 5.

## 2 Methods

This section will outline the used LES model and the treatment of radiation and land-surface interactions, followed by a more detailed description of the bulk microphysics implemented in the Parellized Large-Eddy Simulation Model (PALM) and the extensions made in the scope of the present study.

### 2.1 LES model with embedded radiation and land surface model

In this study the LES model PALM (Maronga et al. 2015; revision 2675 and 3622) was used with additional extensions in the microphysics parameterizations. PALM has been successfully applied to simulate the stable boundary layer (BL) (e.g. during the first intercomparison of LES for stable BL, GABLS, Beare et al., 2006) as well as radiation fog (Maronga and Bosveld, 2017). The model is based on the incompressible Boussinesq-approximated Navier-Stokes equations, and prognostic equations for total water mixing ratio, potential temperature, and subgrid-scale turbulence kinetic energy. PALM is discretized in space using finite differences on a Cartesian grid. For the non-resolved eddies a 1.5-order flux-gradient subgrid closure scheme after Deardorff (1980) is applied, which includes the solution of an additional prognostic equation for the subgrid-scale TKE. Moreover, the discretization for space and time is done by a fifth-order advection scheme after Wicker and Skamarock (2002) and a third-order Runge-Kutta time-step scheme (Williamson, 1980), respectively. The interested reader is referred to Maronga et al. (2015) for a detailed description of the PALM model.

In order to account for radiative effects on fog and the Earth's surface energy balance, the radiation code RRTMG (Clough et al., 2005) has been recently coupled to PALM, running as an independent single column model for each vertical column of the LES domain. RRTMG calculates the radiative fluxes (shortwave and longwave) for each grid volume while considering profiles of pressure, temperature, humidity, liquid water, the droplet number concentration ($n_c$), and effective droplet radius ($r_{eff}$). Compared to the precursor study of Maronga and Bosveld (2017), improvements in the microphysics parameterization

introduced in the scope of the present study allow a more realistic calculation of the fog's radiation budget as $n_c$ is now represented as a prognostic quantity instead of the previously fixed value specified by the user. This involves an improved calculation of $r_{eff}$, entering RRTMG, and which is given as

$$r_{eff} = \left( \frac{3\, q_l\, \rho}{4\pi\, n_c \rho_l} \right)^{\frac{1}{3}} \exp(\log(\sigma_g)^2), \tag{1}$$

where $q_l$ is the liquid water mixing ratio, $\rho$ the density of air, $\rho_l$ being density of water and $\sigma_g$=1.3 being the geometric standard deviation of the droplet distribution. The effective droplet radius is the main interface between the optical properties of the cloud and the radiation model RRTMG. Note, that 3D radiation effects of the cloud are not implemented in this approach, which, however, could affect the fog development at the lateral edges during formation and dissipation phases when no homogeneous fog layer is present. As radiation calculations traditionally require enormous computational time, the radiation code is called
at fixed intervals on the order of 1 min.

Moreover, PALM's land surface model (LSM) is used to calculate the surface fluxes of sensible and latent heat. The LSM consists of multi-layer soil model, predicting soil temperature and soil moisture, as well as a solver for the energy balance of the Earth's surface using a resistance parameterization. The implementation is based on the ECMWF-IFS land surface parametrization (H-TESSEL) and its adaptation in the DALES model (Heus et al., 2010). A description of the LSM and a
validation of the model system for radiation fog is given in Maronga and Bosveld (2017).

## 2.2   Bulk microphysics

As a part of this study, the two-moment microphysics scheme of Seifert and Beheng (2001; 2006) implemented in PALM, basically only predicting the rain droplet number concentration ($n_r$) and cloud water mixing ($q_r$), was extended by prognostic equations for $n_c$ and cloud water mixing ratio ($q_c$). The scheme of Seifert and Beheng (2001; 2006) is based on the separation
of the cloud and rain droplet scale by using a radius threshold of 40 μm. This separation is mainly used for parameterizing coagulation processes by assuming different distribution functions for cloud and rain droplets. However, as collision and coalescence are weak in fog due to small average droplet radii, the production of rain droplets is negligible. Consequently, only the number concentration and mixing ratio of droplets (containing all liquid water and thus abbreviated with $q_l$ here) are considered in the following. The budgets of the cloud water mixing ratio and number concentration are given by

$$\frac{\partial q_l}{\partial t} = -\frac{\partial u_i q_l}{\partial x_i} + \left( \frac{\partial q_l}{\partial t} \right)_{activ} + \left( \frac{\partial q_l}{\partial t} \right)_{cond} - \left( \frac{\partial q_l}{\partial t} \right)_{auto} - \left( \frac{\partial q_l}{\partial t} \right)_{accr} - \left( \frac{\partial q_l}{\partial t} \right)_{sedi}, \tag{2}$$

$$\frac{\partial n_c}{\partial t} = -\frac{\partial u_i n_c}{\partial x_i} + \left( \frac{\partial n_c}{\partial t} \right)_{activ} - \left( \frac{\partial n_c}{\partial t} \right)_{evap} - \left( \frac{\partial n_c}{\partial t} \right)_{auto} - \left( \frac{\partial n_c}{\partial t} \right)_{accr} - \left( \frac{\partial n_c}{\partial t} \right)_{sedi}. \tag{3}$$

The terms on the right-hand side represent the decrease or increase by advection, activation, diffusional growth, autoconversion, accretion, and sedimentation (from left to right). Following Ackerman et al. (2009), cloud water sedimentation is parameterized assuming that droplets are having a log-normal distribution and are following a Stokes regime. This results in a sedimentation

flux of

$$F_{q_l} = k_F \left( \frac{4}{3} \pi \rho_l n_c \right)^{-\frac{2}{3}} (\rho q_l)^{\frac{5}{3}} \exp(5 \ln^2 \sigma_g), \tag{4}$$

with the parameter $k_F = 1.2 \cdot 10^8 \, \text{m}^{-1}\text{s}^{-1}$ (Geoffroy et al., 2010). The main focus of this paper is to study the effect of different microphysical parameterizations of activation and condensation processes on microphysical and macroscopic properties of radiation fog. Those different activation and supersaturation parameterizations will be discussed in the following.

### 2.2.1 Activation

It is well known that the aerosol distribution and the activation process are of great importance for the life cycle of fog (e.g. Gultepe et al., 2007). The amount of activated aerosols determines the number concentration of droplets within the fog, which, in turn, has a significant influence on radiation through optical thickness as well as on sedimentation and consequently affects macroscopic properties of the fog, like for instance its vertical extent. For these reasons, a sophisticated treatment of the activation process is an essential prerequisite for the simulation of radiation fog. Several activation parameterizations for bulk microphysics models have been proposed in literature. In this work, three of these activation schemes were compared with each other in order to quantify their effect on the development of a radiation fog event. The schemes considered in this scope are the activation scheme of Twomey (1959) which was used, e.g., by Bott and Trautmann (2002) to simulate radiation fog, the scheme of Cohard et al. (1998) (used by e.g. Stolaki et al., 2015; Mazoyer et al., 2017) and the one by Khvorostyanov and Curry (2006). The latter two represent an empirical and analytically extension of Twomeys scheme, respectively. Consequently, these parameterizations are frequently termed Twomey-type parameterizations that have the following form:

$$N_{CCN}(s) = N_0 s^k, \tag{5}$$

where $N_{CCN}$ are the number of activated cloud condensation nuclei (CCN), $N_0$ and $k$ are parameters depending on the aerosol distribution, and $s$ is the supersaturation. The three parameterizations considered in the present study are variations of Eq. 5 differing in mathematical complexity:

1. **Twomey (1959):** The power law expression (Eq. 5) is well known and has been used for decades to estimate the number of activated aerosol for a given air mass in dependence of the supersaturation. A weakness of this approach is that the parameters $N_0$ and $k$ are usually assumed to be constant and are not directly linked to the microphysical properties. Furthermore, this relationship creates an unbounded number of CCN at high supersaturations.

2. **Cohard et al. (1998):** extended Twomey's power law expression by using a more realistic four-parameter CCN activation spectrum as shaped by the physiochemical properties of the accumulation mode. Although an extension to the multi-modal representation of an aerosol spectrum would be possible, all relevant aerosols that are activated in typical supersaturations within clouds and especially fog are represented in the accumulation mode (Cohard et al., 1998; Stolaki et al., 2015). Following Cohard et al. (1998) and Cohard and Pinty (2000) the activated CCN number concentration is

expressed by

$$N_{\text{CCN}}(s) = Cs^k \cdot F\left(\mu, \frac{k}{2}, \frac{k}{2}+1; \beta s^2\right), \tag{6}$$

where $C$ is proportional to the total number concentration of CCN that is activated when supersaturation $s$ tends to infinity. Beside $k$, the parameters $\mu$, and $\beta$ are adjustable shape parameters associated with the characteristics of the aerosol size spectrum such as geometric mean radius and the geometric standard deviation as well as with chemical composition and solubility of the aerosols. Thus, in contrast to the original Twomey approach, the effect of physiochemical properties on the aerosol spectrum are taken into account.

3. **Khvorostyanov and Curry (2006):** have found an analytical solution to express the activation spectrum using Koehler theory. Therein, it is assumed that the dry aerosol spectrum follows a log-normal size distribution of aerosol $f_d$:

$$f_d = \frac{dN_{\text{a}}}{dr_{\text{d}}} = \frac{N_{\text{t}}}{\sqrt{2\pi}\ln\sigma_{\text{d}}r_{\text{d}}}\exp\left[-\frac{\ln^2(r_{\text{d}}/r_{\text{d0}})}{2\ln^2\sigma_{\text{d}}}\right]. \tag{7}$$

Here, $r_{\text{d}}$ is the dry aerosol radius, $N_{\text{t}}$ the total number of aerosols, $\sigma_d$ is the dispersion of the dry aerosol spectrum, and $r_{\text{d0}}$ is the mean radius of the dry particles. The number of activated CCN as a function of supersaturation $s$ is then given by

$$N_{\text{CCN}}(s) = \frac{N_{\text{t}}}{2}[1 - \text{erf}(u)]; \qquad u = \frac{\ln(s_0/s)}{\sqrt{2}\ln\sigma_{\text{s}}}, \tag{8}$$

where erf is the Gaussian error function, and

$$s_0 = r_{\text{d0}}^{-(1+\beta)}\left(\frac{4A_{\text{K}}^3}{27b}\right)^{1/2}, \qquad \sigma_s = \sigma_{\text{d}}^{1+\beta}. \tag{9}$$

In this case, $A_{\text{K}}$ is the Kelvin parameter and $b$ and $\beta$ depend on the chemical composition and physical properties of the soluble part of the dry aerosol.

Since prognostic equations were neither considered for the aerosols nor their sources and sinks, a fixed aerosol background concentration was prescribed by setting parameters $N_0$, $C$ and $N_{\text{t}}$ for the three activation schemes. The different nomenclature of the aerosol background concentration is based on the nomenclature used in the original literature.

The activation rate is then calculated as

$$\left(\frac{\partial n_{\text{c}}}{\partial t}\right)_{activ} = \max\left(\frac{N_{\text{CCN}} - n_{\text{c}}}{\Delta t}, 0\right), \tag{10}$$

where $n_{\text{c}}$ is the number of previously activated aerosols that are assumed to be equal to the number of pre-existing droplets and $\Delta t$ is the length of the model time step. Note that this method does not take into account reduction of CCN. However, this error can be neglected since processes as aerosol washout and dry deposition are of minor importance for radiation fog. For all activation schemes it is assumed that every activated CCN becomes a droplet with an initial radius of 1 μm. This results in a change of liquid water, which is considered by the condensation scheme and is described in the next section. Furthermore, we performed a sensitivity study with initial radii of 0.5 μm to 2 μm, which showed that the choice of the initial radius had no impact on the results (not shown). This is consistent with the findings of Khairoutdinov and Kogan (2000) and Morrison and Grabowski (2007).

### 2.2.2 Condensation and supersaturation calculation

The representation of diffusional growth, evaporation, and calculating the underlying supersaturation (which is the main driver for activation) is one of the fundamental tasks of cloud physics. Three different methods have been evaluated and widely discussed in the scientific community. Namely these are the saturation adjustment scheme, the diagnostic scheme, where the supersaturation is diagnosed by the prognostic fields of temperature and water vapor, and a prognostic method for calculating the supersaturation following e.g. Clark (1973); Morrison and Grabowski (2007); Lebo et al. (2012). Basically, the supersaturation is given by $s = q_v/q_s - 1$, while the absolute supersaturation (or water vapor surplus) is defined as $\delta = q_v - q_s$, where $q_v$ is the water vapor mixing ratio and $q_s$ is the saturation mixing ratio. In the following, these three methods are briefly reviewed.

1. **Saturation adjustment:** In many microphysical models, a saturation adjustment scheme is applied. The basic idea of this scheme is that all supersaturation is removed within one model time step and supersaturations are thus neglected. Saturation adjustment thus potentially leads to excessive condensation. Despite the many years of application of this scheme, its impact on microphysical processes is discussed controversially (e.g. Morrison and Grabowski, 2008; Thouron et al., 2012; Lebo et al., 2012). Saturation adjustment might hence especially be a source of error in fog simulations where very small time steps are used due to small grid spacings as already discussed. Using the saturation adjustment scheme, $q_l$ represents a diagnostic value calculated by means of

$$q_l = \max(0, q - q_r - q_s), \tag{11}$$

   where $q$ is the total water mixing ratio. The saturation mixing ratio, which is a function of temperature, is approximated in a first step by

$$q_s(T_l) = \frac{R_d}{R_v} \frac{e_s(T_l)}{p - e_s(T_l)}, \tag{12}$$

   where $T_l$ is the liquid water temperature and $p$ is pressure. $R_d$ and $R_v$ are the specific gas constants for dry air and water vapor, respectively. For the saturation vapor pressure ($e_s$) an empirical relationship of Bougeault (1981) is used. In a second step, $q_s$ is corrected using a first-order Taylor series expansion of $q_s$:

$$q_s(T) = q_s(T_l) \frac{1 + \gamma q}{1 + \gamma q_s(T_l)}, \tag{13}$$

   with

$$\gamma = \frac{L_v}{R_v c_p T_l^2}, \tag{14}$$

   where $c_p$ is the specific heat of dry air at constant pressure and $L_v$ is the latent heat of vaporization. As aforementioned, in each model time step, all supersaturation is converted into liquid water or, in subsaturated regions, the liquid water is reduced until saturation. In order to use this scheme with aerosol activation parameterizations, it is necessary to estimate the supersaturation (see Eq. 5). This can be achieved for the activation scheme of Cohard et al. (1998) following

e.g. Thouron et al. (2012); Mazoyer et al. (2017); Zhang et al. (2014) and directly translating into a droplet number concentration by

$$s^{k+2} \cdot F\left(\mu, k/2, k/2+1, -\beta s\right) = \frac{\left(\phi_1 w + \phi_3 \frac{dT}{dt}\big|_{\text{rad}}\right)^{\frac{3}{2}}}{2kC\pi\rho_l\phi_2 B(\frac{k}{2}, \frac{3}{2})}, \tag{15}$$

where $\phi_1$, $\phi_2$ and $\phi_3$ are functions of temperature and pressure and given in Cohard et al. (1998) and Zhang et al. (2014). $w$ is the vertical velocity and $B$ the beta function.

2. **Diagnostic supersaturation calculation:** Supersaturation is calculated diagnostically from $q_{\text{v}}$ and temperature $T$ (from which $q_{\text{s}}$ can be derived). However, since it is assumed that the supersaturation is kept constant during one model time step, the diagnostic approach requires a very small model time step of

$$\Delta t \leq 2\tau, \tag{16}$$

due to stability reasons (Árnason and Brown Jr, 1971). Here, $\tau$ is the supersaturation relaxation time which is approximated by

$$\tau \approx (4\pi D n_c \bar{r})^{-1}, \tag{17}$$

where $\bar{r}$ is the average droplet radius, and $D$ the diffusivity of water vapor in air. Due to the low dynamic time step in the present study imposed by the Courant-Friedrichs-Lewy criterion (on the order of 0.1 s), however, the condensation time criterion is fulfilled, and no additional time step decrease is needed. The rate of cloud water change due to condensation or evaporation is given by

$$\left(\frac{\partial q_l}{\partial t}\right)_{\text{cond}} = \frac{4\pi G(T,p)\rho_{\text{w}}}{\rho_{\text{a}}} s \int_0^\infty r f(r)dr \tag{18}$$

$$= \frac{4\pi G(T,p)\rho_{\text{w}}}{\rho_{\text{a}}} s r_{\text{c}} \tag{19}$$

where $r_{\text{c}}$ is the integral radius and $G = \frac{1}{F_{\text{K}}+F_{\text{D}}}$ included the thermal conduction and the diffusion of water vapor (Khairoutdinov and Kogan, 2000). The density ratio of liquid water and the solute is given by $\rho_{\text{w}}/\rho_{\text{a}}$.

3. **Prognostic supersaturation:** The prognostic approach, which was first introduced by Clark (1973), includes an additional prognostic equation for the absolute supersaturation. Even though this requires solving one more prognostic equation, it mitigates the problem of spurious cloud-edge supersaturations and prevent inaccurate supersaturation caused by small errors in the advection of heat and moisture (Morrison and Grabowski, 2007; Grabowski and Morrison, 2008; Thouron et al., 2012).

The temporal change of $\delta$ is given by

$$\frac{\partial \delta}{\partial t} - \frac{1}{\rho}\nabla \cdot (u\,\rho\delta) = A - \frac{\delta}{\tau}, \tag{20}$$

with $A$ described by

$$A = -q_s \frac{\rho g w}{p - e_s} - \frac{dq_s}{dT} \cdot \left[ \frac{g w}{c_p} + \left( \frac{dT}{dt} \right)_{\text{rad}} \right], \tag{21}$$

with $g$ being gravitational acceleration. The supersaturation relaxation time is given in Eq. 17. The second term on the left hand side of Eq. 20 describes the change of the absolute supersaturation due to advection, while the right hand side considers changes of $\delta$ due to to changes in pressure, adiabatic compression/expansion, and radiative effects (from left to right). By doing so, the predicted supersaturation is used for determining the number of activated droplets as well as the condensation and evaporation processes. Note that here the absolute supersaturation is taken, as using $s$ would involve more terms and is more complex to solve (Morrison and Grabowski, 2007).

## 3 Case description and model setup

The simulations performed in the present study are based on an observed deep fog event during the night from 22 to 23 March 2011 at the Cabauw Experimental Site for Atmospheric Research (CESAR). The fog case is described in detail in Boers et al. (2013) and was used as validation case for PALM byMaronga and Bosveld (2017). The CESAR site is dominated by rural grassland landscape and, although it is relatively close to the sea, continental aerosol conditions are commonly observed and are characterized by agricultural processes (Mensah et al., 2012).

The fog initially formed at midnight (as a thin near-surface layer), induced by radiative cooling, which also produced a strong inversion with a temperature gradient of 6 K between the surface and the 200 m tower-level. In the following, the fog layer began to develop: at 0300 UTC the fog had a vertical extension of less than 20 m, then deepened rapidly to 80 m, and reaching 140 m depth at 0600 UTC. At 0300 UTC, also the visibility had reduced to less than 100 m. After sunset (around 0545 UTC) a further invigoration close to the ground was suppressed and after 0800 UTC the fog starts quickly evaporate due to direct solar heating of the surface. For details, see Boers et al. (2013).

The model was initialized as described in the precursor study of Maronga and Bosveld (2017). Profiles of temperature and humidity (see Fig. 1) were derived from the CESAR 200 m-tower and used as initial profiles in PALM. A geostrophic wind of $5.5 \, \text{m s}^{-1}$ was prescribed based on the observed value at Cabauw at 0000 UTC.

The land surface model was initialized with short grassland as surface type and four soil model layers at the depths of 0.07 m, 0.28 m, 1.0 m and 2.89 m. The measured surface layer temperatures were interpolated to the respective levels, resulting in temperatures of 279.54, 279.60, 279.16, and 279.16 K for soil layers one to four, respectively. Furthermore, the initial soil moisture was set to the value at field capacity ($0.491 \, \text{m}^3\text{m}^{-3}$), which reflects the very wet soil and low water table in the Cabauw area. Moreover, the roughness length for momentum was prescribed to 0.15 m. Note that Maronga and Bosveld (2017) discussed that this value appears to be a little high given the season and wind direction. This does not play an important role for the present study, however, as we will not focus on direct comparison against observational data from Cabauw.

All simulations start at 0000 UTC, before fog formation, and end at 1015 UTC on the next morning after the fog layer has fully dissipated. Precursor runs are conducted for additional 25 min using the initial state at 0000 UTC, but without radiation

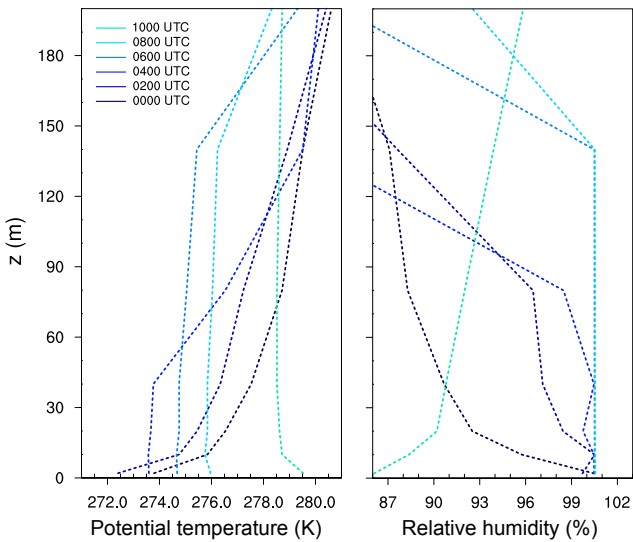

**Figure 1.** Profiles of potential temperature and relative humidity at different times as observed at Cabauw.

scheme and LSM in order to allow the development of turbulence in model without introducing feedback during that time (see Maronga and Bosveld, 2017).

Based on sensitivity studies of Maronga and Bosveld (2017), a grid spacing of $\Delta = 1\,\mathrm{m}$ was adopted for all simulations, with a model domain size of 768 x 768 x 384 grid points in $x$-, $y$-, and $z$-direction, respectively. Cyclic conditions were used at the lateral boundaries. A sponge layer was used starting at a height of 344 m in order to prevent gravity waves from being reflected at the top boundary of the model.

Table 1 gives an overview of the simulation cases. All cases were initialized with (identical) continental aerosol conditions. Case SAT represents a reference run with no activation scheme and thus a prescribed constant value of $n_\mathrm{c} = 150\,\mathrm{cm}^{-3}$ (estimated from simulations of Boers et al., 2013). This case represents the same setup to the one described in Maronga and Bosveld (2017) except of modifications concerning the aerosol environment as outlined below. Condensation processes were here treated with the saturation adjustment scheme (Seifert et al., 2006). In order to evaluate the influence of saturation adjustment in a one-moment microphyiscs scheme on the development of radiation fog, identical assumptions were made in case DIA and PRG, except that diffusional growth was calculated with the diagnostic and prognostic method, respectively (see section 2.2.2).

Moreover, as small differences in supersaturation can effect the number of activated droplets significantly the impact of different methods for calculating supersaturation on CCN activation is investigated in a two-moment microphysics approach (see section 4.2.2). Therefore, the simulation N2SAT, N2DIA and N2PRG were compared to each other. In all three cases the activation scheme of Cohard et al. (1998) is used and initialized as described below.

Furthermore, cases N1DIA-N3DIA used the activation schemes described in chapter 2.2.1. To ensure comparability between the different schemes, all of them were initialized with a continental aerosol background described in Cohard et al. (1998),

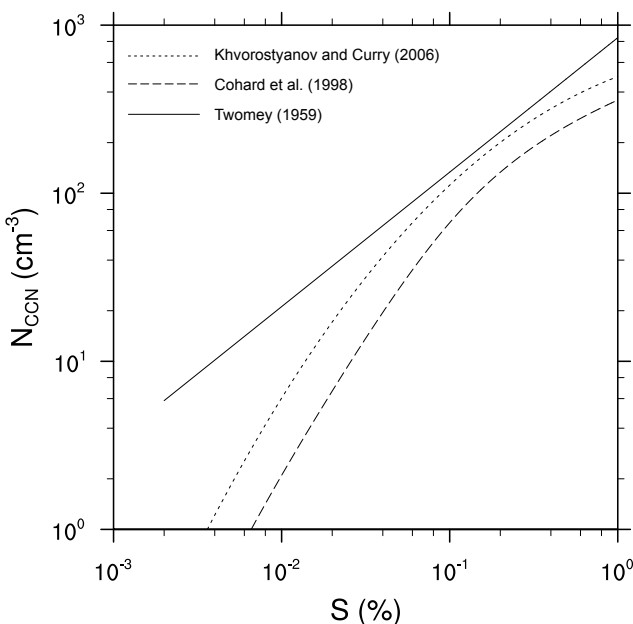

**Figure 2.** Activation spectrum for three different activation schemes of Twomey (1959), Cohard et al. (1998) and Khvorostyanov and Curry (2006) for a typical continental aerosol environment.

which is characterized by an aerosol with the chemical composition of ammonium sulfate [$(NH_4)_2SO_4$], a background aerosol concentration of $842\,cm^{-3}$, a mean dry aerosol radius of $r_{d0} = 0.0218\,\mu m$, and a dispersion parameter of the dry aerosol spectrum of $\sigma_d = 3.19$. For the Twomey activation scheme this results in $N_0 = 842\,cm^{-3}$ and $k = 0.8$ which is a typical value for the exponent for continental air masses (e.g. Pruppacher and Klett, 1997, pages 289 et seq.). The Twomey activation

5   scheme does not allow for taking aerosol properties into account. In contrast, the activation scheme of Cohard et al. (1998) requires the parameters $C$, $k$, $\beta$ and $\mu$ to be derived from the aerosol properties. Here, values of $C = 2.1986 \cdot 10^6\,cm^{-3}$, $k = 3.251$, $\beta = 621.689$ and $\mu = 2.589$ were used as described in Cohard and Pinty (2000). Finally, the activation scheme of Khvorostyanov and Curry (2006) can directly consider the aerosol properties, which are prescribed as aforementioned. Using those different parameterizations resulting in different activation spectra, which are shown in Fig. 2. One can see, that especially

10   the CCN concentration is changed by using these different methods, such that this part of the study is equivalent to sensitivity study of different CCN concentration but realized by using different coexisting parameterizations.

**Table 1.** Overview of conducted simulations. The droplet number concentration $n_c$ is only prescribed for simulations without activation scheme. In the simulations N1DIA-N3DIA $n_c$ is a prognostic quantity and thus variable in time and space. The aerosol background concentration is abbreviated with $N_{a,tot}$, and used to initialize the activation schemes. Note for the scheme after Cohard et al. (1998) a conversion to the parameter $C$ must be applied, while for both other activation schemes this value is directly used to prescribe $N_0$ and $N_t$, respectively.

| # | Simulation | Activation scheme | $n_c$ [cm$^{-3}$] | $N_{a,tot}$ [cm$^{-3}$] | Condensation scheme |
|---|---|---|---|---|---|
| 1 | SAT | none | 150 | none | saturation adjustment |
| 2 | DIA | none | 150 | none | diagnostic |
| 3 | PRG | none | 150 | none | prognostic |
| 4 | N2SAT | Cohard et al. (1998) | not fixed | 842 | saturation adjustment |
| 5 | N2DIA | Cohard et al. (1998) | not fixed | 842 | diagnostic |
| 6 | N2PRG | Cohard et al. (1998) | not fixed | 842 | prognostic |
| 7 | N1DIA | Twomey (1959) | not fixed | 842 | diagnostic |
| 8 | N3DIA | Khvorostyanov and Curry (2006) | not fixed | 842 | diagnostic |

## 4 Results

### 4.1 General fog life cycle and macrostructure

The reference case SAT is conducted with a constant droplet number concentration of $n_c = 150\,\text{cm}^{-3}$. The deepening of the fog layer can be seen in Fig. 3, which shows the profiles of the potential temperature, relative humidity and liquid water mixing ratio at different times.

The fog onset is at 0055 UTC, defined by a visibility below 1000 m and a relative humidity of 100%. In the following the fog layer deepens and extends to a top of approximately 20 m at 0200 UTC. However, at this point the stratification of the layer is still stable with a temperature gradient of 6 K between the surface and the fog top. The persistent radiative cooling of the surface and the fog layer leads to a further vertical development of the fog, which is accompanied with a regime transition from stable to convective conditions within the fog layer (see Fig. 3a). This starts as soon as the fog layer begins to become optically thick (at 0330 UTC), and when radiative cooling at the fog top becomes the dominant process, creating a top-down convective boundary layer. The highest liquid water mixing ratio of $q_l = 0.41\,\text{g}\,\text{kg}^{-1}$ is achieved at 0600 UTC at a height of 60 m (see Fig. 3c), while the fog layer in total reaches the maximum one hour later at 0700 UTC. The lifting of the fog, which is defined by a non-cloudy near-surface layer ($q_l \leq 0.01\,\text{g}\,\text{kg}^{-1}$), occurs at 0845 UTC. At 1130 UTC the fog is completely dissipated.

### 4.2 Influence of different supersaturation calculation

In this section we discuss the influence of three different method considering supersaturation. Namely these are (as afore-mentioned) saturation adjustment, a diagnostic supersaturation calculation and a prognostic method. In the first subsection a one-moment microphysic scheme is used and the impact of the different supersaturation methods is limited to the effect of

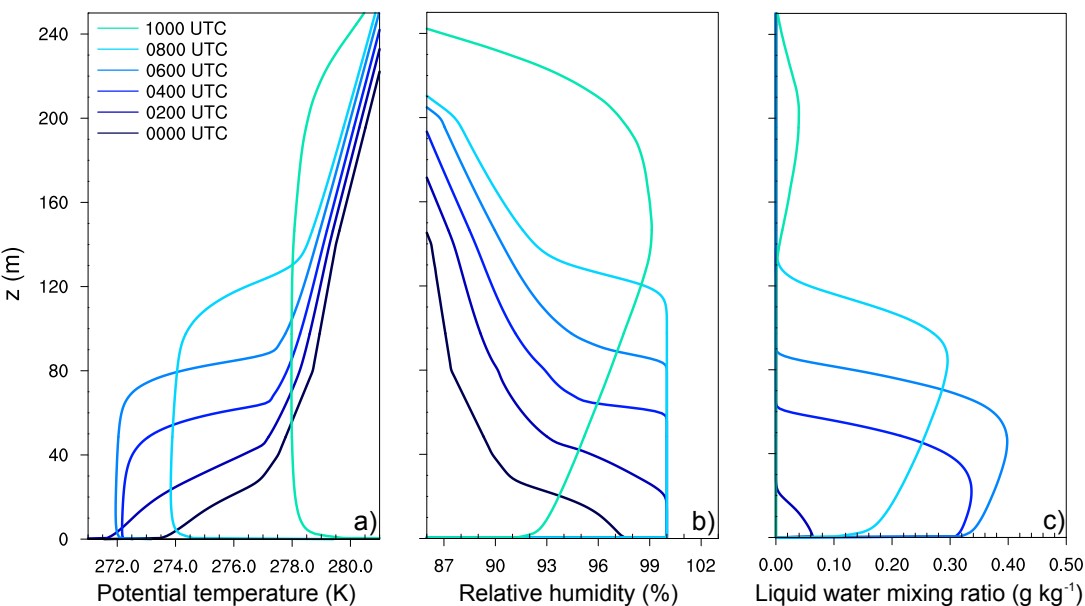

**Figure 3.** Profiles of potential temperature (a), relative humidity (b) and liquid water mixing ratio (c) at different times for the reference case REF.

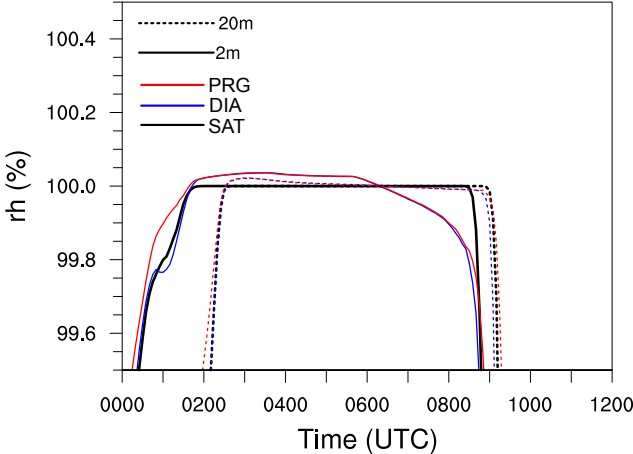

**Figure 4.** Time series of horizontal-averaged relative humidity/supersaturation at height levels of 2 m (solid) and 20 m (dotted) for different methods in treating the supersaturation calculation.

diffusional growth. In the second part of this study those methods are applied in a two-moment microphysics scheme and considering the effect of such different approaches of supersaturation calculations for activation.

### 4.2.1 One-moment microphysics scheme: impact of supersaturation calculation on diffusional growth

In this section we discuss the error introduced by using saturation adjustment for simulating radiation fog with a one-moment scheme in a LES. For this, we compare three simulations with identical setup (cases SAT, DIA, and PRG), which differs only in the way how supersaturation is calculated and consequently the amount of condensed or evaporated liquid water. To isolate this effect, activation is neglected in all cases and $n_c$ is set to a constant value of $150\,\mathrm{cm}^{-3}$ (a typical value in fog layers). The effect on different supersaturations driving the diabatic process of activation is discussed in section 4.2.2. As mentioned before the time step is roughly $0.1\,\mathrm{s}$, which is more than one order of magnitude smaller than the allowed values of 2 - 5 s for assuming saturation adjustment (Thouron et al., 2012). The present case hence is an ideal environment evaluating the error introduced by using saturation adjustment and by keeping all other parameters fixed.

Figure 4 shows time series of the horizontally-averaged saturation (supersaturation) for cases SAT, DIA and PRG at selected heights close to the surface. In all cases saturation occurs simultaneously around 0120 UTC. In case SAT, relative humidity does not exceed 100% due to its limitation by saturation adjustment, while in case DIA and PRG average supersaturations of 0.05% are reached at a height of $2\,\mathrm{m}$, which corresponds to typical values within fog (Hammer et al., 2014; Mazoyer et al., 2019; Boutle et al., 2018).

For cases DIA and PRG starting from 0615 UTC (in $2\,\mathrm{m}$ height) and 0715 UTC (in $20\,\mathrm{m}$ height), supersaturations are removed and the air becomes subsaturated (on average). This is in contrast with case SAT, where the saturation adjustment approach keeps the relative humidity at 100% as long as liquid water is present (i.e. until the fog has dissipated). Around 0600 UTC, which is shortly after sunrise, relative humidity drops rapidly in PRG and DIA as a direct consequence of direct solar heating of the surface and the near-surface air, preventing further supersaturation at these heights. While we cannot clearly identify the lifting of the fog in case DIA and PRG (due to the limited humidity range displayed), we note that for case SAT we can identify lifting times as a decrease of relative humidity around 0845 UTC at $2\,\mathrm{m}$ height and around 0910 UTC at $20\,\mathrm{m}$ height.

Beside this inherent difference in relative humidity, the general time marks (formation, lifting, dissipation, defined by Maronga and Bosveld, 2017) of the fog layer are identical for cases SAT, DIA and PRG.

Figure 5 shows the liquid water path (LWP) for all cases. Differences in the LWP appear between 0400 UTC and 1100 UTC and do not exceed 1% (lower values for cases DIA and PRG), indicating that the choice of the condensation scheme does not affect the total water content of the simulated fog layer.

It can be summarized that, although the assumptions of saturation adjustment are not valid for the simulation of fog when using a very small time step, the mean liquid water content is not changed by more than 1% and the general fog structure is not altered when using a one-moment microphysics and neglecting supersaturation. This is probably due to the very small supersaturation that is not strong enough to generate a significant change in the effective droplet radius, and which could possibly lead to stronger sedimentation or higher radiative cooling rates.

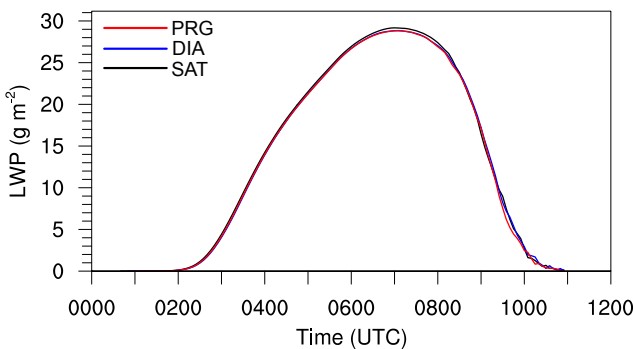

**Figure 5.** Time series of liquid water path (LWP) for cases using saturation adjustment, the diagnostic approach and a prognostic method for the diffusional growth.

### 4.2.2 Two-moment microphysics scheme: impact of supersaturation calculation on CCN activation

Even though different methods for calculating supersaturation which interacts with the diffusional growth are not strong enough to generate any noteworthy differences by using a one-moment microphysics (considering a constant value for $n_c$) the impact of different methods modelling supersaturation on CCN activation by using a two-moment microphysics might be significant.

Figure 6 shows the LWP for simulations applying the activation scheme of Cohard et al. (1998) in conjunction with the usage of saturation adjustment (N2SAT), the diagnostic scheme (N2DIA), and the prognostic scheme (N2PRG) for calculating supersaturations. It can be seen that the prognostic and diagnostic methods produce similar LWP values. However, for case N2SAT the LWP is nearly 70% higher than for the other two cases. In Fig. 7 profiles of the liquid water mixing ratio (left) and droplet number concentration (right) are shown. From that figure it can be seen that in case of N2SAT both the fog

height as well as the liquid water mixing ratios within the layer are higher than in N2DIA and N2PRG, respectively. However, small differences in $q_l$ can also be found between N2DIA and N2PRG (e.g. at 0600 UTC in the second third of the fog layer). This is explained by slightly higher values for the number concentration in case of N2DIA than in N2PRG. However, both are at approximately $75\,\mathrm{cm}^{-3}$ at 0600 UTC. In contrast, in simulation N2SAT a number concentration of $120\,\mathrm{cm}^{-3}$ to $150\,\mathrm{cm}^{-3}$ (at the top) is observed, which is about 60%-100% higher in comparison to N2DIA and N2PRG. These differences

can be explained by the different methods for calculating the supersaturation, since activation is the main process altering the droplet number concentration. Therefore, we can implicitly derive from the droplet number concentration that the predicted and diagnosed supersaturations using the prognostic and diagnostic method are similar. These differences between N2SAT and N2DIA/N2PRG are, however, in good agreement with values reported for a stratocumulus case by Thouron et al. (2012). Their Fig. 2 shows that the number concentration of the diagnostic and prognostic method were also similar and the case with

saturation adjustment overestimated the supersaturation and therefore the droplet number concentration. As the fog droplet number concentration has a crucial feedback on the overall LWP of the fog layer, the times of lifting, and the time of its dissipation, the reported differences in $n_c$ are significant regarding the accurate modelling and prediction of fog. The reason

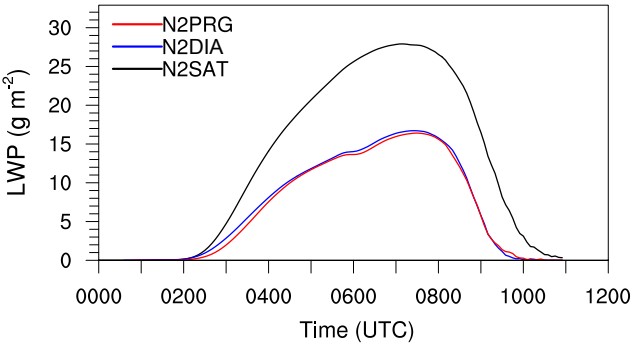

**Figure 6.** Time series of LWP for simulations using saturation adjustment (N2SAT, black), the diagnostic scheme (N2DIA, blue) and the prognostic method (N2PRG, red). All cases uses the activation scheme of Cohard et al. (1998).

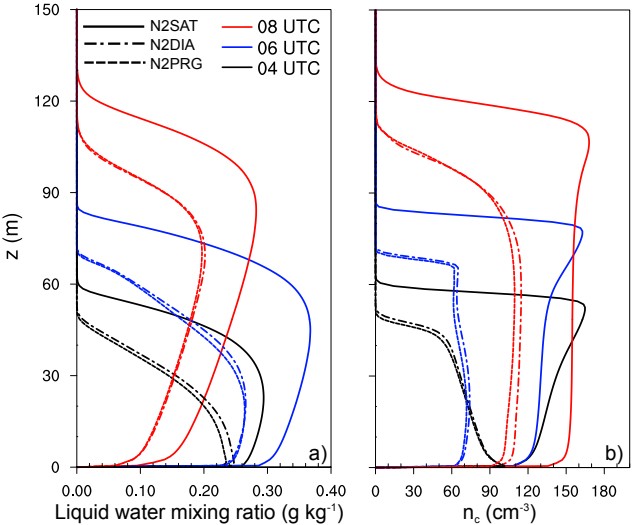

**Figure 7.** Profiles for liquid water mixing ratio (a) and droplet number concentration (b) at 0400 UTC, 0600 UTC and 0800 UTC.

why the number concentration is such a critically parameter can be ascribed to their impact on sedimentation and radiative cooling, which is explained in more detail in section 4.4.3.

In order to evaluate the possible effect of the grid spacing, in conjunction with different methods for calculating the super-saturation, on CCN activation, we repeated each of the cases N2SAT, N2DIA, and N2PRG with two coarser grid spacings of 2 m and 4 m. The general effect of the grid spacing on the temporal development and structure of radiation fog is discussed in detail in Maronga and Bosveld (2017). In this section, we will focus only on changes in LWP due to different supersaturation calculations at different spatial model resolutions. For isolating the effect of the grid spacing, all simulations with a coarser grid spacing were carried out with the same time step of 0.125 s, which corresponds to the average time step of the simulations at highest grid spacing of 1 m. In this way, effects of different time steps induced by different grid spacings, could be eliminated.

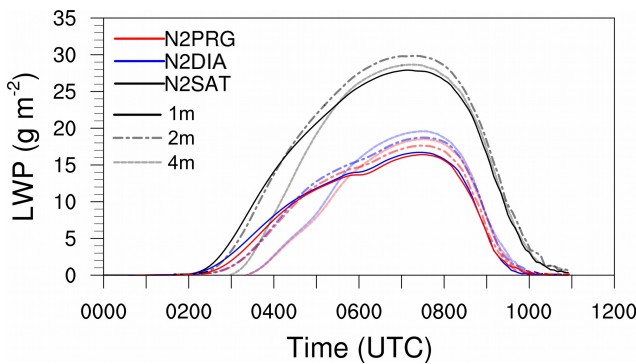

**Figure 8.** As Fig. 6 but also 2 m (dot-dashed) and 4 m (dashed).

Fig. 8 shows the LWP for all grid sensitivity runs. First of all, note that for 1 m grid spacing, the results reflect the results shown in Fig. 6 and discussed above (i.e. significantly higher LWP for case N2SAT than for cases N2DIA and N2PRG. Moreover, Fig. 8 reveals that these results are somewhat sensitive to changes in the grid spacing. For all cases we observe a tendency towards higher LWP values with increasing grid spacing, at least for cases N2DIA and N2PRG. These difference are, however, not larger than $4\,\mathrm{g\,m^{-2}}$ and thus significantly smaller than the observed differences found between the different methods to calculate supersaturation. Note, however, that the relative change in LWP with grid spacing is higher for case N2DIA than for case N2PRG. Quantitatively speaking, in case of 1 m grid spacing the relative difference of the LWP is 2.1% between N2DIA and N2PRG during the mature phase while for the case with a grid spacing of 4 m it reaches 8.1%. This might be explained by the fact that the diagnostic scheme is very sensitive to small errors (e.g. induced by the numerical advection) in the temperature and humidity fields (e.g. Morrison and Grabowski, 2008; Thouron et al., 2012). A coarser spatial resolution here can lead to larger error introduced by spurious supersaturation. We thus suppose that the increased differences (see Fig. 8) by larger grid spacings are induced by spurious supersaturation, which affect the CCN activation and hence influence the LWP of the fog layer.

Furthermore, we note that coarser grid spacings lead to later fog formation time, which is in agreement with Maronga and Bosveld (2017) and which can be ascribed to under-resolved turbulence near the surface at coarse grids.

In summary, we can thus conclude that the sensitivity to changes in the grid spacing is rather small, but it might imply differences in the LWP of the simulated fog layer of up to $4\,\mathrm{g\,m^{-2}}$.

### 4.3 Two-moment microphysics scheme: comparison of different activation parameterizations

In numerous previous studies, the influence of aerosols and the activation process on the life cycle of fog was investigated (e.g. Bott, 1991; Stolaki et al., 2015; Maalick et al., 2016; Zhang et al., 2014; Boutle et al., 2018). Although all three activation schemes outlined in section 2.2.1 are comparable power law parameterizations that are initialized with identical aerosol spectra, the effect on simulations of radiation fog is still unknown. Because changes in $n_c$ due to different activation schemes have a considerable effect on the life cycle of fog, we might consider that even small differences in $n_c$ might alter simulated fog layers

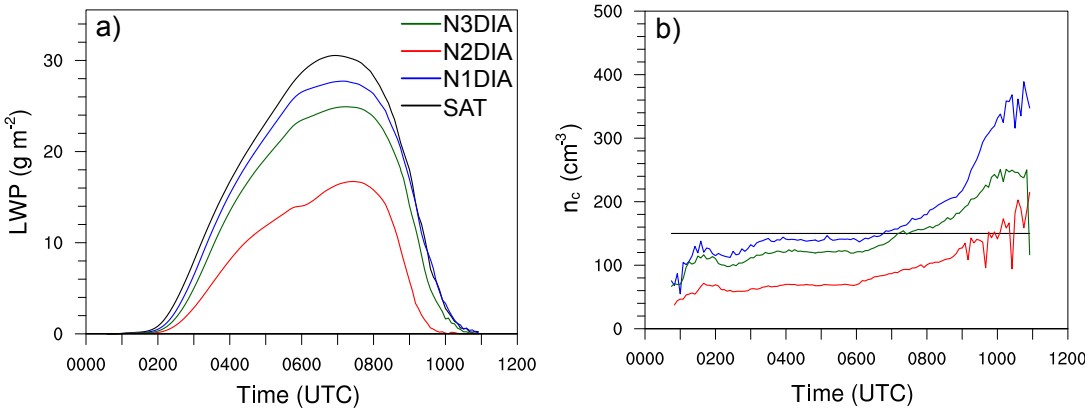

**Figure 9.** Time series of LWP and $n_c$ (as a horizontal and vertical average of the fog layer) for the reference and N1DIA-N3DIA case.

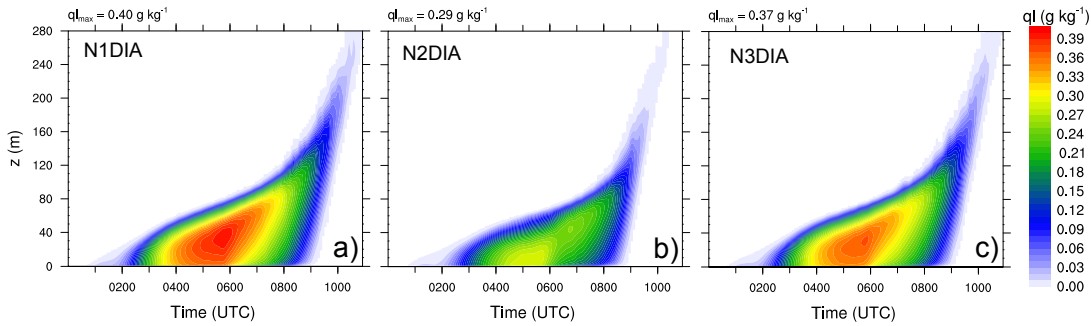

**Figure 10.** Height-time cross sections for the liquid water mixing ratio for N1DIA-N3DIA.

significantly. This part of the study can be regarded as a sensitivity study of different CCN concentrations realized by applying different activation schemes, which is illustrated also in Fig. 2. However, from a model user's perspective, such a sensitivity is of great importance as CCN concentrations are usually difficult (case studies) or even impossible (forecasting) to obtain and model results thus might highly depend on the chosen activation parameterization.

## 4.4 LWP and $n_c$

Time series of the LWP for the reference run (case SAT) and the three different cases (N1DIA - N3DIA) are shown in Fig. 9a. The highest LWP occurs for case SAT which also shows the highest $n_c$ during the formation and mature phase in comparison with the other simulations (see 9b). The time series of $n_c$ shown in 9b (representing runs with the three different aerosol activation parameterization schemes, see Tab.1) reveal that, depending on the parameterization used, the a shift in $n_c$ towards smaller or larger values is found. The quantitative differences in the number of activated aerosol by using the different activation schemes is due to a slightly different activation spectrum (see Fig. 2). A linear relationship between LWP and $n_c$ can be found: a higher $n_c$ leads to higher LWP, which is in agreement to other studies as Boutle et al. (e.g 2018). In principle, a similar

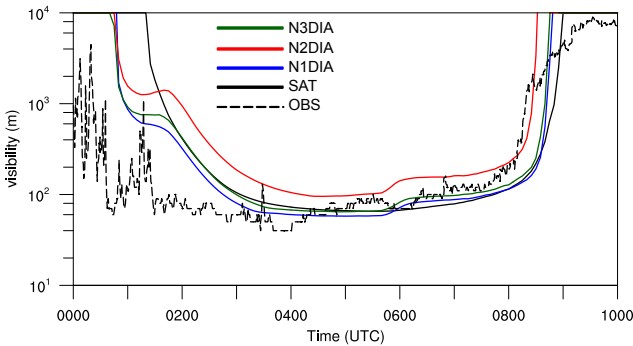

**Figure 11.** Time series of simulated visibility in 2 m height. Observations from Cabauw (dashed lines) were added for illustration only.

qualitative development of $n_c$ can be observed. While $n_c$ increases during fog formation (with a local maximum with values between 70 and 140 $\mathrm{cm}^{-3}$), it remains nearly constant during the mature phase of the fog (values between 65 and 145 $\mathrm{cm}^{-3}$). We will see later see that activation here happens mostly at the top of the fog, but due to vertical mixing in the convective fog layer, cloud droplets are evenly distributed over a large vertical domain. Furthermore, the mixing layer is increasing in

time so that there is no net change of the (averaged) $n_c$ in the fog layer. As soon as the sun rises and the fog layers start to lift and turns into a stratocumulus cloud, all cases show a strong increase in $n_c$. This increase can be explained by stronger supersaturations induced by thermal updrafts in the developing surface-driven convective boundary layer due to surface heating by solar radiation. Moreover, we note that while the qualitative course of $n_c$ is similar for all cases, the choice of the activation algorithm has an impact on the number of activated aerosols and thus on the strength of the fog-layer, e.g. illustrated in Fig. 10

via $q_l$. This is due to the radiation effect of the droplets. The number of droplets to which a certain amount of liquid water is distributed plays an important role: the larger the number of droplets, the larger is the radiation-effective surface and the higher also the optical thickness. As a result the cooling rate in a fog with many small droplets is increased, allowing more water vapor to condense and the fog to grow stronger. By the same token, sedimentation also depends on the droplet radius and plays a major role for fog development. This will be further discussed below.

**4.4.1 Visibility**

In Fig. 11 the simulated visibility for the cases N1DIA-N3DIA in 2 m height together with the observed values at Cabauw (for illustration only). Visibility is calculated from the LES data following Gultepe et al. (2006) as

$$vis = \frac{1002}{(n_c \, \rho \, q_l)^{0.6473}}, \tag{22}$$

with $n_c$ and $q_l$ given in units of $\mathrm{cm}^{-3}$ and $\mathrm{gm}^{-3}$, respectively. This visibility estimation thus significantly depends on the

droplet number concentration and the liquid water content. Unlike in the first part of this paper, analyzing visibility estimations from the simulations might illuminate the capability of LES to predict visibility. Fig. 11 reveals that visibility follows the same general temporal developed for all cases, with a rapid decrease at fog formation, deepening, and dissipation; with minimum

**Table 2.** Table of fog's life cycle time marks.

| Simulation | Onset | Maximum | Lifting | Dissipation |
|------------|----------|----------|----------|-------------|
| N1DIA | 0025 UTC | 0510 UTC | 0810 UTC | 1005 UTC |
| N2DIA | 0050 UTC | 0425 UTC | 0755 UTC | 0910 UTC |
| N3DIA | 0025 UTC | 0515 UTC | 0810 UTC | 0950 UTC |

values around 100 m (which is close to the observed values). We also see noteworthy differences, particularly shortly before 0200 UTC (before fog deepening) at around 0545 UTC (shortly after sunrise). For both time marks, case N1DIA - N3DIA display sudden increases in visibility, due to an fast decrease of $n_c$ in 2 m height; and which are not reproduced by case SAT, as $n_c$ is fixed value in this case. The sudden increase in visibility around 0045 UTC in the observations is possibly related to this

process. Also, the time marks of formation and dissipation vary. For cases N1DIA - N3DIA the formation time is significantly advanced compared to case SAT, while dissipation time only shows a small tendency towards earlier times, at least for N1DIA and N3DIA. Case N2DIA displays a different behavior, with a later fog formation and higher visibility and accordingly earlier dissipation time. This is in line with the findings discussed above (i.e. a much weaker fog layer that, as a direct consequence, can dissipate much faster). Otherwise, all cases display almost identical visibility as soon as the fog has deepened.

**4.4.2   Time marks of the fog life cycle**

The effect of the different droplet concentration (induced by the usage of different activation schemes) on the time marks of the fog life cycle is summarized in Tab. 2. While N1DIA and N3DIA have similar time marks, N2DIA stands out and show a delayed onset by 25 min, while the maximum liquid water mixing ratio is reached 45 min earlier than in the other cases. Also lifting and dissipation are affected and occurred 15 min and 40 min (with respect to simulation N3DIA) earlier. This is

due to a lesser absolute liquid water mixing ratio which evaporates faster by the incoming solar radiation. Therefore, it can be concluded that the use of different activation schemes (if they change the droplet number concentration) has an effect on the time marks on the life cycle as well as on the fog height and the amount of liquid water within the fog layer.

**4.4.3   Budgets of liquid water and droplet number concentration**

In this section we will analyze the budgets of liquid water and droplet number concentration in physical terms. As in the

preceding section, we will use the cases with different activation parameterizations, since they provide us a range of different CCN concentrations. Figure 12a shows the profiles of the liquid water mixing ratio at 0400 UTC, 0600 UTC, and 0800 UTC, i.e at different times during the mature phase of the fog. A detailed analysis of budgets at other stages of the life cycle of the fog is beyond the scope of this paper. The maximum $q_l$ in the fog layer is reached at approximately 0600 UTC at a height of 60 m. Afterwards a further vertical growth of the fog can be observed, where no further increase in liquid water takes places as a result

of larger vertical extent of the mixing layer and due to rising temperatures after sunrise. Moreover, Fig. 12b,c show the liquid water budget during the mature phase of the fog at 0600 UTC, when the fog was fully developed. Almost all three cases show

identical values for condensation rates in the lowest part of the fog layer, with values being in the same order as the evaporation rates, so that the net gain in this region appears to be small (see Fig. 12b). However, the N2DIA case (with the lowest $n_c$) exhibits a generally lower absolute evaporation rate compared to both other cases, which can be attributed to the slightly higher mean values of the relative humidity (not shown) than in N1DIA and N3DIA. In the upper part of the fog layer, higher values of

the condensation rate are observed (especially for N1DIA and N3DIA) with a concurrent decrease in evaporation rates, leading to differently strong deepening of the fog layer. At a height of approximately 80 m a maximum of the evaporation rates can be observed, representing the presence of subsaturated regions in this height and the top of the fog. Larger differences can be observed in the sedimentation rates: First and foremost the sedimentation is proportional to the liquid water mixing ratio (see also Eq. 4). The strength of sedimentation also depends on the mean radius of the droplets, which increases with decreasing

number of activated drops. Here, a lower $n_c$ for a given amount of liquid water leads to a higher mean radius, compared to a higher $n_c$ where the same amount of water is distributed to more drops, decreasing the mean radius. Integrated over height all three cases exhibit approximately the same sedimentation rates. Therefore, case N2DIA experiences the strongest loss of liquid water due to sedimentation (in relative terms). Moreover, Fig. 12c shows that sedimentation partially counteracts the gains caused by condensation at the upper edge of the fog. The net advection transports liquid water from the second third of

the fog layer (position of the maximum) to higher levels. It can be summarized that all terms contribute significantly to the net change of the liquid water mixing ratio, illustrating that all microphysical processes deserve a proper modelling for radiation fog. In the mature phase, however, sedimentation plays a key role, showing the highest values for the individual tendencies. As a result liquid water is slowly and constantly removed from the fog layer. These findings are in good agreement with previous investigations by Bott (1991).

The sum of all tendencies, which is shown in Fig. 12d, is the height-dependent change of the liquid water. Also here it can be seen that in the lower 50 m the net tendency is negative, while in higher levels we observe a positive tendency, so that the fog continues growing vertically, while the liquid water content within the fog layer decreases.

Figure 13a additionally shows the profiles of $n_c$. We note that the profiles of the different cases differ quantitatively but not qualitatively. The stage of the fog can thus be identified in the profiles for all cases: At 0400 UTC highest supersaturations

occur close to the ground due to cooling of the surface and near-surface air, leading to high activation rates and therefore high $n_c$ near the surface (not shown). At 0600 UTC a well-mixed layer has developed that is driven by the radiative cooling from the fog top. While the turbulent mixing leads to a vertical well-mixed $n_c$, we note the maximum at the top, where the radiative cooling induces immense aerosol activation. This is further illustrated in the budget of the $n_c$ in Fig. 13b,c, where instantaneous data at 0600 UTC is shown. Here, we see clearly that aerosol activation at the top of the fog layer is the dominant

process in the mature phase of the fog, while activation near the surface is comparably small. Evaporation of droplets, though small in magnitude, occurs only at the fog top, reflecting upward motions of foggy air penetrating the subsaturated air aloft where droplets then evaporate. Also, we see that both advection and sedimentation rates are much smaller than activation rates, so that the net change in $n_c$ is controlled by the activation near the fog-top during the mature phase of the fog.

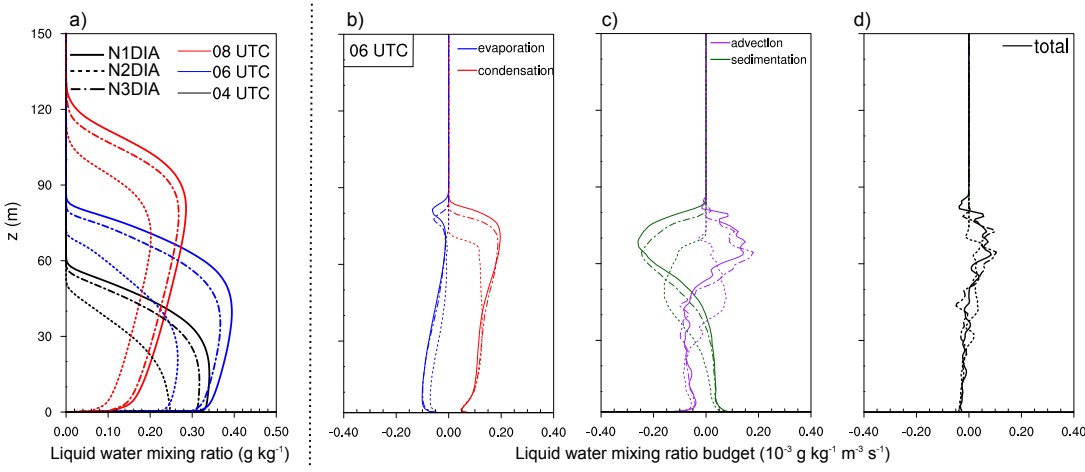

**Figure 12.** Profiles (instantaneously and horizontally averaged) of liquid water mixing ratio at 0400 UTC, 0600 UTC and 0800 UTC and profiles of liquid water budget terms at 0600 UTC.

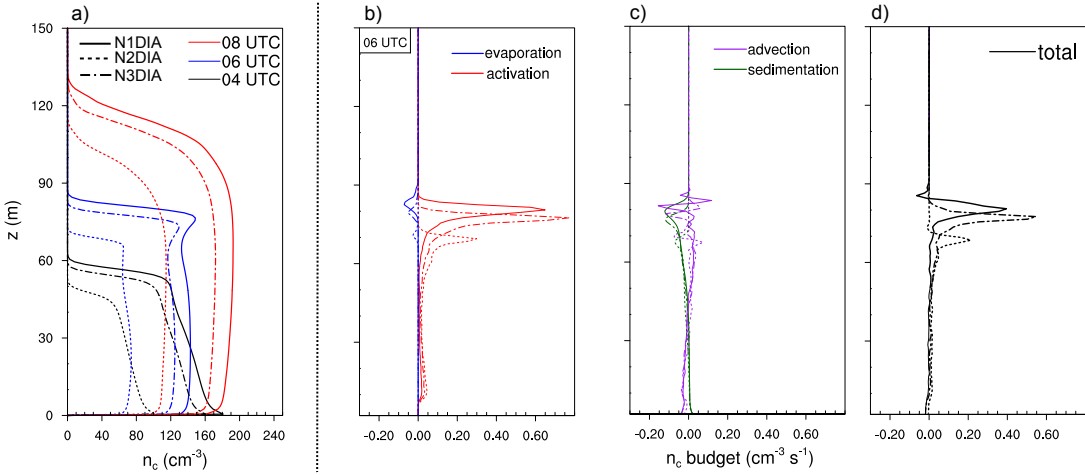

**Figure 13.** Profiles (instantaneously and horizontally averaged) of $n_c$ at 0400 UTC, 0600 UTC and 0800 UTC and profiles of $n_c$ budget terms at 0600 UTC.

## 5 Conclusions

The main objective of this work was to investigate the influence of the choice of the supersaturation calculation and activation parameterizations used in LES models on the life cycle of simulated nocturnal deep radiation fog under typical continental aerosol conditions. For this purpose we performed a series of LES runs based on a typical deep fog event as observed at

5    Cabauw (Netherlands).

In the main part of this study we applied a two-moment microphysics scheme with an activation parameterization of Cohard et al. (1998) and investigated the influence of three different (but commonly used) supersaturation calculation methods, i.e. saturation adjustment, a diagnostic method, and a prognostic method, on the life cycle and LWP of the simulated fog event. From the results we found that in case of saturation adjustment nearly 60% higher droplet number concentration are produced in comparison to simulation with the diagnostic or prognostic method. This results in a more than 70% higher LWP for the saturation adjustment case and a later occurrence of lifting and dissipation of the fog layer. An explanation for such differences between the schemes can be found in the general assumptions made within the methods. As saturation adjustment assumes that the complete water vapor surplus is removed within one time step, the supersaturation used for activation must be parameterized. In agreement with Thouron et al. (2012) we found that those values are higher as in the other cases, which leads to great feedback of the fog layer. Moreover, we found that the diagnostic method and the prognostic method yield similar results. However, in a grid spacing sensitivity study we observed that the relative differences between the prognostic and diagnostic approach increase as the spatial resolution decrease. We assume that this is due to larger errors of spurious supersaturations which lead to an overestimation of activation in the diagnostic case. This in turn effect the sedimentation velocity as well as the effective radius and hence the radiative cooling, which results in higher values for the LWP.

In a further test, using a one-moment microphysics scheme, we compared the possible error introduced by using saturation adjustment in comparison with an diagnostic and prognostic method for calculating the supersaturation for diffusional growth, i.e. neglecting activation and prescribing a constant droplet number concentration. With this assumptions we were able to isolate the error introduced by saturation adjustment on condensation and evaporation. However, the results showed that, although the model time step was inappropriate for the assumptions made during saturation adjustment, the differences in LWP are at most 1% and the general life cycle is not affected. This could be attributed to the fact that the typical supersaturations in fog are in the range of a few tenths of a percent, and the resulting absolute differences are too small to induce a further influence on dynamics, microphysics or radiation. This result implies that saturation adjustment is an acceptable method if no activation parameterization is available (with simultaneous consideration that the latter is highly recommended).

In a second part of our study, the effect of different activation schemes of Twomey (1959), Cohard et al. (1998) and Khvorostyanov and Curry (2006) on the simulated fog life cycle was investigated. Even though these parameterizations appear to be rather similar, our results indicate that the resulting number of activated aerosols (and consequently the number of droplets), known to be a crucial parameter for the fog development, can differ significantly. However, it must be mentioned that these differences are attributed to the fact that the CCN concentration is different for the investigated schemes. This part of the study can thus also be understood as a sensitivity study for different CCN concentrations realized by the usage of different activation schemes.

In order to get a deeper insight into the spatial and temporal development of deep radiation fog, we performed an additional analysis of budgets $n_c$ and $q_l$ during the mature phase of the fog for simulations with different aerosol activation parameterizations. We found that gain of liquid water is dominated by condensational growth throughout the fog layer with a maximum at the top of the fog layer (due to longwave radiative cooling) and by significant sedimentation of fog droplets from upper levels towards lower levels, while only little liquid water is lost by sedimentation (to the ground) and evaporation. The fact that the

simulated cases displays significant differences in the fog strength could be traced back to the differences in the condensational growth at the fog top, induced by different activation of CCN. For $n_c$, our simulations indeed indicate that activation is the dominant process, located in a narrow height level, while all other processes (i.e. evaporation, advection, and sedimentation) were found to be comparably small. The amount of generated liquid water thus is a direct consequence of the strength of the

activation process and is thus related to the number of CCN and accordingly the activation parameterization used in the model.

In summary, the present study indicates that the choice of the used supersaturation calculation can be a key factor for the simulation of radiation fog. In agreement with Thouron et al. (2012) we recommend to use the prognostic approach to calculate the supersaturation for fog layer in case of a two-moment microphysics considering activation. Therewith, the effect of spurious cloud edge supersaturation is mitigated and too large activation rates are omitted. Further, the choice of the chosen

activation scheme has a noticeable impact on the number concentration of CCN and hence on the LWP and fog layer depth. However, we have no means to give advice which activation parameterization performs best. In order to give a more educated recommendation here, we would need observational data of size distributions from aerosol and fog droplets.

In order to overcome the remaining limitations of the present study that are related to microphysical parameterizations, we are currently working on a follow-up study in which we are revisiting this particular fog case using a Lagrangian particle-

based approach to simulate the microphysics of droplets. This will allow for explicitly simulating the development of the 3D droplet size distribution in the fog layer (e.g. Shima et al., 2009). This approach will also allow to resolve all relevant microphysical processes such as activation and diffusional growth directly, instead of parameterizing them. As such simulations are computationally very expensive, only a very limited number of simulations are feasible at the moment, so that most future numerical investigations will - as in the present work - rely on bulk microphysics parameterizations. Based on the results using

the Lagrangian approach, however, we hope to be able to give an educated recommendation on the best choice for such bulk parameterizations.

*Code availability.* The PALM model used in this study (revision 2675 and revision 3622) is publicly available on http://palm-model.org/trac/browser/palm?rev=2675 and http://palm-model.org/trac/browser/palm?rev=3622, respectively. For analysis, the model has been extended and additional analysis tools have been developed. The extended code, as well as the used Job-Setups and the used PALM source code are publicly

available on https://doi.org/10.25835/0067929. All questions concerning the code-extension will be answered from the authors on request.

*Author contributions.* The numerical experiments were jointly designed by the authors. JS implemented the microphysics parameterizations, conducted the simulations and performed the data analysis. Results were jointly discussed. JS prepared the manuscript, with significant contributions by BM.

*Competing interests.* The authors declare that they have no conflict of interest.

*Acknowledgements.* This work has been funded by the German Research Foundation (DFG) under Grant MA 6383/1-1, which is greatly acknowledged. All simulations have been carried out on the Cray XC-40 systems of the North-German Supercomputing Alliance (HLRN, https://www.hlrn.de/).

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
