# Peer review of "Large-eddy simulation of radiation fog with comprehensive two-moment bulk microphysics: impact of different aerosol activation and condensation parameterizations"

_Atmospheric Chemistry and Physics, 2018_

## Referee Comment (RC1) · Anonymous Referee #1 · 20 Nov 2018

**Review of the paper entitled**
**"Large-eddy simulation of radiation fog with comprehensive two-moment bulk microphysics: impact of different aerosol activation and condensation parameterizations"**

by Johannes Schwenkel and Bjorn Maronga

**manuscript number ACP-2018-1139**

This paper addresses the difficult topic to evaluate the influence of cloud microphysical parameterizations on large-eddy simulation of radiation fog. The results are based on one case of deep fog observed at Cabauw (Netherlands). The subject of the manuscript is interesting as radiation fogs are not well known, and particularly the influence of microphysical processes on the fog life cycle. However, I think that some revisions will be helpful to make this paper clearest.

1. **clarify the effect of microphysical parameterizations on the fog life cycle** :
   Following Fig. 8, the microphysical parameterizations used do not modify the fog onset, the time when fog becomes optically thick, the lifting time of fog and the time when fog is completely dissipated. However, it is very difficult to evaluate precisely these parameters from Fig. 8. I think that a table summarizing these 4 times, crucial in the fog life cycle (onset, transition into optically thick fog, lifting time and complete dissipation), would be helpful to evaluate the impact of the parameterizations used. Could you please add this table and discuss the impact of microphysical parameterizations on these parameters? Please elaborate.

2. **effect of microphysical parameterizations on visibility at ground level** :
   Your simulations demonstrate that the microphysical parameterizations mainly impact the microphysical properties of the fog layer (liquid water mixing ratio and LWP). These parameters ($q_l$ and $n_c$) have a significant impact on the diagnosed visibility. Could you please discuss the impact of the microphysical parameterizations used on the diagnosed visibility at ground level? Is this impact significant?

Or is this impact of the same magnitude than uncertainties due to visibility diagnostic? Please elaborate.

3. **effect of aerosol** :
   Your tests are done for a background aerosol concentration of $842cm^{-3}$ and for a given aerosol chemical composition. What is the impact of this hypothesis on your results? Are your results also valid in a highly polluted atmosphere (e.g. observation made during WIFEX), or in an atmosphere with low aerosol concentration? Please elaborate.

4. **shallow fog / deep fog** Are your findings also true for shallow fog (with thermal inversion at ground level)? The dynamical processes between shallow and deep (mature) fog are strongly different. And consequently, the impact of microphysical parameterizations could be very different during the fog life cycle (due to difference in supersaturation magnitude). Could you please clarify the sensitivity of microphysical parameterizations depending on the fog type (ie shallow vs deep fog)?

5. Stolaki et al. (2015) use 1D model. She does not use 2D LES. Please modify (p2 l2).

6. Figure 6b, 6c, 6d, 9c and 10c are very hard to read (too many curves on the same plot). Could you please try to improve these figures?

---

## Referee Comment (RC2) · Anonymous Referee #2 · 19 Dec 2018

The manuscript presents a study of condensation and activation parametrizations for a LES of radiation fog. This is an interesting topic as most of LES of fog now use 2-moment microphysical schemes and also produce an overestimation of cloud concentration and mass. Therefore these questions of activation are central. The relevance of saturation adjustment for LES has been raised by Thouron et al. (2012) for stratocumulus and Lebo et al. (2012) for deep convective clouds. Since these studies, it is the first time that this question is dealing with fog. So this study could be an original contribution to the modelling community. But the study suffers from a lot of weaknesses and

is not convincing. Therefore it misses the objective. Whilst the topic is interesting, and could be ultimately worthy of publication, I feel major modifications to the manuscript are required, and substantial inputs are necessary before publication.

My major concerns areÂă:

- The case is an observed fog event, but you never show observation so there is no reference. Therefore you cannot say that liquid water content is overestimated in some configuration.

- You draw conclusions with only one case. For instance 6.9Âă% just corresponds to one case and you generalize this result to characterize the impact of adjustment saturation for fog (in the abstract/conclusion). In the same way, for the sensitivity of the time step, you claim that you test a larger time step without showing the result, and you state that the effect is not negligible. This is not scientific and admissible. A broad range of time steps needs to be compared. Additionally, what is the sensitivity to the spatial resolutionÂă?

- The objective to evaluate the impact of saturation adjustment was promising but disappointing as you do not compare explicit vs saturation adjustment for 2 moment microphysical scheme, despite the fact that 2 moment microphysical schemes are the most frequently used in LES of fog. At least a N0 test with Twomey or Cohard and saturation adjustment needs to be added, to be compared to N1 or N2. Moreover a more complete study of this topic would include a pseudo-prognostic approach of supersaturation (Thouron et al., 2012).

- The comparison of different activation parametrizations (4.3) is reduced to a sensitivity test to the CCN concentration, and contributes nothing new. Why have you not chosen more equivalent activation properties, for instance if the 3 curves pass by the same point S=0.1Âă% NCCN=100 cm-3 (Fig.A1) in order to compare the 3 parametrizationsÂă? Because the 3 activation schemes present different curvatures according to S, and this point is not discussed.

- There are a lot of inaccuracies.

More specificallyÂă:

1. The introduction has been neglected and does not raise the scientific questions. The fact that most of LES of fogs produce an overestimation of cloud concentration and mass is one argument to justify this study (See Mazoyer et al., 2017). 2. p2Âă: Stolaki et al. (2015) used 1D simulations 3. p2 l 7Âă: What is SalsaÂă? ReferenceÂă? 4. p 2 l 11Âă: Mazoyer et al. (2017) needs to be added 5. p 2 l 20Âă: Thouron et al. (2012) is the first paper raising the question of how relevant the saturation adjustment is for LES of clouds. The paper draws extensively on Thouron et al. (2012) but it is not sufficiently referenced in different parts. 6. p2 l31Âă: What does revision 2675 meanÂă? 7. p 3Âă: some information about PALM is missingÂă: What are the numerical schemes usedÂă? Is the turbulence scheme 1D or 3DÂă(does it parametrize horizontal turbulent fluxes) ? More importantÂă: what are the parametrizations for the computation of cloud optical propertiesÂă? 8. p 7Âă: The explicit supersaturation calculation corresponds to the scheme B in Thouron et al. (2012) (diagnostic of supersaturation). They have shown that this method is very sensitive to small errors in temperature and mixing ratio. Spurious values of supersaturation have a significant impact on CCN activation. They showed that it also overestimates CCN activation at the top. All this information should be recalled as well as the reference. 9. P7 line 15-17 is not clear. Could you improve the explanation if you want to justify that a pseudo-prognostic approach is not interesting or necessary. 10. Tab 1 and Part 4Âă: please add and analyze a new test N0 with Twomey or Cohard and saturation adjustment. 11. Fig 3Âă: you say ÂńÂăheight averagedÂăÂż and then 2m and 20m. So whatÂă? 12. Fig.4Âă: do time marks refer to C1 or REFÂă? 13. P11 l 4Âă: why are the time steps in the pluralÂă? Can you also explain shortly why they are so smallÂă? 14. P 12 l 17Âă: it is C1 minus REF, isn't itÂă? 15. P12 l 21-22Âă: How are these higher liquid mixing ratios producedÂă? 16. P 12 l 27Âă: Again why is the time step approximatedÂă? 17. P12 l 26-35Âă: This paragraph is not acceptable as you conclude on a sensitivity of the time

step without showing any result. 18. P13 l 4Âă: what is the reference to say that liquid water is overestimatedÂă? Why do not you use the observed valueÂă? 19. Fig 7Âă: nc is a 3D field. So is it a vertical and horizontal average, or is it for the first vertical levelÂă? 20. P 14 l 21Âă: as it is the explicit method, why do you take care of maximum supersaturationÂă? 21. What is new from Fig. 9 and 10Âă? 22. p 16Âă: Could you conclude that the radiation impact of nc is more important than in the sedimentation processÂă? 23. Fig 9Âă: it would be better to put the total tendency in b than in c, as profiles are too intermingled in c. 24. Fig 10Âă: Deactivation means evaporationÂă?

MisspellingÂă:

- p1 l 20Âă: aerosols - p2 l 9Âă: as as - p12 l 21Âă: diminishes - p14 l 18Âă: is→ are - p 15 l 16Âă: shows

Please also note the supplement to this comment: https://www.atmos-chem-phys-discuss.net/acp-2018-1139/acp-2018-1139-RC2-supplement.pdf

---

## Author Comment (AC1) · 20 Feb 2019

**Review of the paper entitled**
**"Large-eddy simulation of radiation fog with comprehensive two-moment bulk microphysics: impact of different aerosol activation and condensation parameterizations"**

**by Johannes Schwenkel and Bjorn Maronga**

**RC1:** This paper addresses the difficult topic to evaluate the influence of cloud microphysical parameterizations on large-eddy simulation of radiation fog. The results are based on one case of deep fog observed at Cabauw (Netherlands). The subject of the manuscript is interesting as radiation fogs are not well known, and particularly the influence of microphysical processes on the fog life cycle. However, I think that some revisions will be helpful to make this paper clearest.

**Author's answer:** First of all, we would like to thank the reviewer for the detailed and constructive feedback. In particular, the suggestions on the visibility and time marks of the fog cycle were added and are now discussed in the revised manuscript. Furthermore, the other points of criticism regarding the aerosol concentration and the difference between shallow fog and deep fog were taken into account and the manuscript was adapted accordingly. With the help of these comments, it was possible to contribute to a significant improvement in the work and to clarify the research.

**RC1: 1. clarify the effect of microphysical parameterizations on the fog life cycle :**
Following Fig. 8, the microphysical parameterizations used do not modify the fog onset, the time when fog becomes optically thick, the lifting time of fog and the time when fog is completely dissipated. However, it is very difficult to evaluate precisely these parameters from Fig. 8. I think that a table summarizing these 4 times, crucial in the fog life cycle (onset, transition into optically thick fog, lifting time and complete dissipation), would be helpful to evaluate the impact of the parameterizations used. Could you please add this table and discuss the impact of microphysical parameterizations on these parameters? Please elaborate.
**Author's answer:** We agree with this objection that a table is the method of choice for displaying these parameters. In the revised manuscript the table is provided in section 4.3 and discussed in the following.
**Modification(p19, l1):** The effect of the different activation schemes on the time of the fog life cycle is summarized in Tab. 2. The largest differences occur for simulation N2EXP in comparison to N1EXP and N3EXP. The onset is delayed by 25 min, while the maximum liquid water mixing ratio is reached 45 min earlier, than in the other cases. Also lifting and dissipation are affected and occurred 15 min and 40 min (with respect to simulation N3EXP) earlier. This is due to a lesser absolute liquid water mixing ratio which is more easily evaporated by the incoming solar radiation. Therefore, it can be concluded that the use of different activation schemes (if they change the droplet number concentration) has an effect on the time marks on the life cycle, even though the general shape stays untouched

**RC1: 2. effect of microphysical parameterizations on visibility at ground level :**
Your simulations demonstrate that the microphysical parameterizations mainly impact the microphysical properties of the fog layer (liquid water mixing ratio and LWP). These parameters (ql and nc) have a significant impact on the diagnosed visibility. Could you please discuss the impact of the microphysical parameterizations used on the diagnosed visibility at ground level? Is this impact significant? Or is this impact of the same magnitude than uncertainties due to visibility diagnostic? Please elaborate.
**Author's answer:** This is a good suggestion, since the visibility was a measured quantity during CESAR. We have added time series of the simulated and measured visibility (even though it was not our aim to represent a specific fog case as close as possible to observations) in the revised version.

**Modification(p2, l2):** In Fig. 9 the simulated visibility for the cases N1-N3 in 2 m height as well as the observed value is shown. For the simulation the visibility is calculated by

$$vis = 1002/(n_c \, \rho \, q_l)^{0.6473},$$

following Gultepe et al. (2006). Here, $n_c$ and $q_l$ must be given in units of $cm^{-3}$ and $gm^{-3}$, respectively. Hence, the visibility is significantly affected by the droplet number concentration and the liquid water content. As one can see all simulations reproduce the general trend of the visibility quite well. During the onset of the fog all simulation tend to underestimate the visibility. In the mature phase Simulation N2 exhibit the largest values to the observed visibility, but matches best during the lifting phase. However, it should be mentioned that it was not our goal to mimic one particular fog case.

**RC1: 3. effect of aerosol :**
Your tests are done for a background aerosol concentration of $842 cm-3$ and for a given aerosol chemical composition. What is the impact of this hypothesis on your results? Are your results also valid in a highly polluted atmosphere (e.g. observation made during WIFEX), or in an atmosphere with low aerosol concentration? Please elaborate.

**Author's answer:** It is true, that our simulations shows results for only one aerosol environment. However, these are well-known and frequently investigated conditions. Furthermore, in many of the observed radiation fog events the underlying conditions are similar the chosen aerosol conditions of this study.
Of course, changed aerosol conditions would change the absolute numbers of the errors done with the investigated microphysical parametrizations. However, the qualitative findings will be untouched. The reason why we have not conducted simulations for only quantifying the difference for different aerosol environments is then based on the needed computational resources which are tremendous (54h on 3072 computer cores).
However, we agree that these findings with their concrete numbers are limited to cases with continental aerosol conditions. Due to that we have adapted the manuscript to clarify that it's based on continental aerosol conditions.

**Modification(different passages):** [..] continental aerosol conditions [..]

**RC1: 4. shallow fog / deep fog**
Are your findings also true for shallow fog (with thermal inversion at ground level)? The dynamical processes between shallow and deep (mature) fog are strongly different. And consequently, the impact of microphysical parameterizations could be very different during the fog life cycle (due to difference in supersaturation magnitude). Could you please clarify the sensitivity of microphysical parameterizations depending on the fog type (ie shallow vs deep fog)?

**Author's answer:** This is an interesting objection. However, in current research the focus is more on deep fog events, as these affect our everyday life much more (e.g. dangers for air and car traffic). Moreover, even though the dynamics of shallow fog compared to a deep fog event might be different, it was not our aim to derive universally valid statements for the entire parameter space (as for different aerosol conditions).
Unfortunately, as stated in the previous comment this is with a high-resolved (isotropic grid-spacing of 1m) LES not possible where one simulation requires many computer resources (as mentioned above).
Form the fundamental research point of view, different microphysical parameterizations might also affect shallow fog since the crucial parameter is the supersaturation.
But we agree, to clarify that our statements are especially valid for a deep fog case under typical continental aerosol conditions, and according to that we had adapted our manuscript.

**Modification(at different passages):** [..] deep [..]

**RC1:** 5. Stolaki et al. (2015) use 1D model. She does not use 2D LES. Please modify (p2 l2).
**Author's answer:** This is right. It is corrected in the revised revision.

**Modification(p2, l2):** [..] while using the one-dimensional mode of the MESO-NH model [..]

**RC1:** 6. Figure 6b, 6c, 6d, 9c and 10c are very hard to read (too many curves on the same plot). Could you please try to improve these figures?

**Author's answer:** As also the other reviewer criticized the figures. We modified them, as we separate them to more individual plots.

**Modification(Fig 6,9,10):** We modified the figures 6,9 10.

---

## Author Comment (AC2) · 20 Feb 2019

**Review of the paper acp-2018-1139 « Large-eddy simulation of radiation fog with comprehensive two-moment bulk microphysics: Impact of different aerosol activation and condensation parameterizations» from Johannes Schwenkel and Björn Maronga**

**RC2:** The manuscript presents a study of condensation and activation parametrizations for a LES of radiation fog. This is an interesting topic as most of LES of fog now use 2-moment microphysical schemes and also produce an overestimation of cloud concentration and mass. Therefore these questions of activation are central. The relevance of saturation adjustment for LES has been raised by Thouron et al. (2012) for stratocumulus and Lebo et al. (2012) for deep convective clouds. Since these studies, it is the first time that this question is dealing with fog. So this study could be an original contribution to the modelling community.
But the study suffers from a lot of weaknesses and is not convincing. Therefore it misses the objective. Whilst the topic is interesting, and could be ultimately worthy of publication, I feel major modifications to the manuscript are required, and substantial inputs are necessary before publication.

**Author's answer:** First of all, we would like to thank the reviewer for the detailed and constructive feedback. Especially the high expert competence of the review allowed us to overcome the weaknesses, to extend the study by reasonable points and to focus the scientific result.

**RC2:** The case is an observed fog event, but you never show observation so there is no reference. Therefore you cannot say that liquid water content is overestimated in some configuration.
**Author's answer:** Indeed, the simulated fog case was an observed event during CESAR. A detailed comparison to measurements is given in Maronga and Bosveld, 2017.  However,  the relevant (in terms of our study needed) quantities, such as droplet number concentration,  liquid water mixing ratios and liquid water path were not measured. In our statements, which claim an overestimation of certain configurations, we therefore refer to theoretical considerations.
However, we agree that in some passages that this was not clear enough or not sufficiently proved. Thus, we modified those passages and only make valuations were there are justified.
**Modification(p2, l2):** Rewrite passages, which claimed an overestimation and could not be sufficiently proved.

**RC2:** You draw conclusions with only one case. For instance 6.9 % just corresponds to one case and you generalize this result to characterize the impact of adjustment saturation for fog (in the abstract/conclusion). In the same way, for the sensitivity of the time step, you claim that you test a larger time step without showing the result, and you state that the effect is not negligible. This is not scientific and admissible. A broad range of time steps needs to be compared. Additionally, what is the sensitivity to the spatial resolution?
**Author's answer:** We agree with you general objection that drawing quantitative conclusions by two simulations is not admissible. Due to that we removed the passages were we generalized our results. Further, we labeled our results as that was they are: Findings from high-resolved LES study with (typical) continental aerosol conditions. From that we can only conclude that similar cases might show a similar trend but may differ in their concrete numbers. Moreover, as you suggested in the next comment we added a prognostic approach for simulating supersaturation. Due to carefully double checking we noted a bug in our model code and must repeat one simulation (old C1, renamed in EXP in the revised manuscript). Accordingly, the quantitative results changed but the general qualitative findings remain untouched.
Furthermore, we removed the conclusion that differences getting smaller by a larger time-step, as it was not sufficiently proved and mixed with a comparison to simulations with a different gridspacing. Therefore. the effect was not isolated to the time step.
Due to computational costs ( one simulation requires approximately 48h on 3072 cores on a supercomputer) a broad range of time steps could not be conducted with this setup, since only (due to dynamical stability reasons) a reduction of the time-step is allowed.
However, we added a sensitivity study with a spatial resolution of 4m and 2m in chapter 4.4 as there changes to the grid-spacing should have the strongest effect.
**Modification(chapter 4.2 and 4.4):** As this referee comment involve major modifications we kindly refer to the revised manuscript and the attached manuscript, which highlights all changes in comparison to the first version individually.

**RC2:** The objective to evaluate the impact of saturation adjustment was promising but disappointing as you do not compare explicit vs saturation adjustment for 2 moment microphysical scheme, despite the fact that 2 moment microphysical schemes are the most frequently used in LES of fog. At least a N0 test with Twomey or Cohard and saturation adjustment needs to be added, to be compared to N1 or N2. Moreover a more complete study of this topic would include a pseudo-prognostic approach of supersaturation (Thouron et al., 2012).

**Author's answer:** We decided to follow the reviewer suggestion and added three more high-resolved simulations.
Firstly, we extended the part of the study where the influence of different condensational growth parameterizations are isolated and investigated (in terms of using a 1D-microphysics with fixed number concentration). Here, we also added a prognostic approach for calculating the supersaturation, which drives the strength of the diffusional growth.
Secondly, as the reviewer proposed we added the saturation adjustment case with a activation scheme of Cohard et al., 1998. Moreover, we also applied the same activation scheme by using the prognostic approach for calculating supersaturation. By doing so, we introduced (following Thouron et al., 2012.) a new section where the influence and feedback of different supersaturation calculation on the droplet activation (by using the scheme of Cohard et al., 1998) is discussed.
For that we compared N2EXP to the new simulations N2SAT and N2PRG. The new introduced simulations are summarized in the Table (bold marked). Note, that though these major modifications we decided to rename the simulation to make it more intuitively.

| # | Simulation | Activation Scheme | Nc | Na | Condensation Scheme |
|---|---|---|---|---|---|
| 1 | SAT | None | 150 | none | Saturation adjustment |
| 2 | EXP | None | 150 | none | explicit |
| **3** | **PRG** | **None** | **150** | **none** | **prognostic** |
| 4 | N1EXP | Twomey | Not fixed | 842 | explicit |
| 5 | N2EXP | Cohard et al | Not fixed | 842 | explicit |
| 6 | N3EXP | Khvorostyanov and Curry (2006) | Not fixed | 842 | explicit |
| **7** | **N2SAT** | **Cohard et al.** | **Not fixed** | **842** | **Saturation adjustment** |
| **8** | **N2PRG** | **Cohard et al.** | **Not fixed** | **842** | **Prognostic** |

**Modification(chapter 2.2.2 and chapter 4.4):** [..] As this referee comment involve major modifications we kindly refer to the revised manuscript and the attached manuscript, which highlights all changes in comparison to the first version individually

**RC2:** The comparison of different activation parametrizations (4.3) is reduced to a sensitivity test to the CCN concentration, and contributes nothing new. Why have you not chosen more equivalent activation properties, for instance if the 3 curves pass by the same point S=0.1 % NCCN=100 cm-3 (Fig.A1) in order to compare the 3 parametrizations ? Because the 3 activation schemes present different curvatures according to S, and this point is not discussed.

**Author's answer:** Our idea here was to show the differences between different activation schemes initializing in such a way (by using the in literature described values, see Cohard et al., 1998, Khvorostyanov and Curry, 2006 and Pruppacher et al., 1998 (chapter 2.2) ) that they are describing the same aerosol environment. So basically we didn't change the aerosol concentration, since we leaved this parameter untouched.

However, considering the activation spectra displayed in A1 we agree that there is mainly a offset between the schemes of Cohard et al., 1998 and Khvorostyanov and Curry, 2006. In contrast to that to the Twomey-scheme we see both, an offset as well as a different curvature. As you suggest we could modify the activation spectrum (or more precisely the parameter describing the aerosol environment) in a such a way that they pass by the same point. But again, the overall goal was more from the view of a users, which is maybe interested in which total differences can be produced by using this or that scheme. As this get not clear enough in the first version of the manuscript, we modified the revised version accordingly.

**Modification(Chapter 1):** [..]

**RC2: There are a lot of inaccuracies. More specifically:**

**RC2(1):** The introduction has been neglected and does not raise the scientific questions. The fact that most of LES of fogs produce an overestimation of cloud concentration and mass is one argument to justify this study (See Mazoyer et al., 2017).

**Author's answer:** After major modifications also the introduction was carefully revised. Furthermore, the study of Mazoyer et al., 2017 was added. But more important, the missing scientific question was clarified.

**Modification(Chapter 1):** [..] As Mazoyer et al. (2017) and Boutle et al. (2018) stated that both, LES and NWP models tend to overestimate the liquid water content and the droplet number concentration for radiation fog the following questions are derived from these shortcomings:
(i) Is saturation adjustment appropriate as it crucially violates the assumption of equilibrium? How large is the effect of different supersaturation calculations on diffusional growth?

(ii) What is the impact of different activation schemes on the fog life cycle for a given aerosol environment?

(iii) As the number of activated droplets is essentially determined by the supersaturation, how large is the effect of different supersaturation modeling approaches on aerosol activation and therewith on the strength and life cycle of radiation fog (cf. Thouron et al., 2012)?

In the present paper we will try to answer these questions employing high-resolution LESs based on an observed typical deep fog event with continental aerosol conditions. The paper is organized as follows: Section 2 outlines the methods used, that is the LES modeling framework and the microphysics parameterizations used. Section 3 provide an overview of the simulated cases and model setup, while results are presented in Section 4. Conclusions are given in Section 5.

**RC2(2):** p2 : Stolaki et al. (2015) used 1D simulations
**Author's answer:** This is right. It is corrected in the revised revision.
**Modification(p2, l2):** [..] while using the one-dimensional mode of the MESO-NH model [..]

**RC2(3):** p2 l 7 : What is Salsa? Reference?
**Author's answer:** That was right, an complete reference was missing. SALSA is a sectional module for a size resolved treating for aerosols.
**Modification(p2, l7):** [..] Sectional Aerosol module for Large Scale Applications (SALSA) (Kokkola et al.,2008) [..]

**RC2(4):** p 2 l 11 : Mazoyer et al. (2017) needs to be added
**Author's answer:** Added in the revised manuscript.
**Modification(p2, l8):** [..]Mazoyer et al., 2017 conducted similar to Stolaki et al., 2015 simulation of the ParisFog with the MESO-NH model but using the 3D-LES mode, and focusing on the influence of drag effect on droplet deposition

**RC2(5):** p 2 l 20 : Thouron et al. (2012) is the first paper raising the question of how relevant the saturation adjustment is for LES of clouds. The paper draws extensively on Thouron et al. (2012) but it is not sufficiently referenced in different parts.
**Author's answer:** We agree, and have added this reference in the missing passages.
**Modification(p?, l?):** e.g. p2, l30, p7,l3, p8, l6

**RC2(6):** p2 l31 : What does revision 2675 mean?
**Author's answer:** Our LES-model PALM is maintained with the trac-system. Due to that every change in the model code or corresponding files is explicitly identified with the revision number. With that number it is also possible to get the for this studies used model code from our web page, which is mentioned in the acknowledgments.
**Modification(p2, l31):** None.

**RC2:(7):** p 3 : some information about PALM is missing : What are the numerical schemes used? Is the turbulence scheme 1D or 3D (does it parametrize horizontal turbulent fluxes)? More important : what are the parametrizations for the computation of cloud optical properties?
**Author's answer:** We have added the missing information about PALM and how optical properties of clouds and how they are treated in the radiation model. By doing so we were as short as possible to avoid to lengthen the manuscript, but more important as precise as necessary.
**Modification(p3-4):** [..] PALM is discretized in space using finite differences on a Cartesian grid. For the non resolved eddies a 1.5-order flux-gradient subgrid closure scheme after Deardorff (1980) is applied, which includes the solution of an additional prognostic equation for the subgrid-scale TKE. Moreover, the discretization for space and time is done by a fifth-order advection scheme after Wicker and Skamarock (2002) and a third-order Runge-Kutta time-step scheme (Williamson, 1980), respectively. The interested reader is referred to Maronga et al., 2015 for a detailed description of the PALM model.

[..]This favors an improved calculation of the effective radius, which is calculated by

$$ r_{eff} = \frac{3\,q_l\rho}{4\,\pi\,n_c\rho_l}^{1/3} \exp\left(\log\left(\sigma_g\right)^2\right), $$

where $q_l$ is the liquid water mixing ratio, rho the air density, $\rho_l$ being density of water and $\sigma_g$=1.3 the geometric standard deviation of the droplet distribution. The effective radius is the main interface between the optical properties of the cloud and the radiation model RRTMG. Note, that 3D radiation effects of the cloud are not implemented in this approach, which however could affect the lateral edges.

**RC2(8):** p 7: The explicit supersaturation calculation corresponds to the scheme B in Thouron et al. (2012) (diagnostic of supersaturation). They have shown that this method is very sensitive

to small errors in temperature and mixing ratio. Spurious values of supersaturation have a significant impact on CCN activation. They showed that it also overestimates CCN activation at the top. All this information should be recalled as well as the reference.

**Author's answer:** We agree, and added a prognostic approach for treating supersaturation to our work. This includes a new chapter discussing the effect of different methods for supersaturation calculation on CCN activation.

**Modification(chapter 2.2.2 and chapter 4.4):** [..] As this referee comment involve major modifications we kindly refer to the revised manuscript and the attached manuscript, which highlights all changes in comparison to the first version individually.

**RC2(9):** P7 line 15-17 is not clear. Could you improve the explanation if you want to justify that a pseudo-prognostic approach is not interesting or necessary.

**Author's answer:** Our primary reasons for not using a prognostic approach for solving the supersaturation was that a small grid spacing is the method of choice to migitate the error introduced by spurious cloud edge supersaturations (e.g. Hoffmann, 2016). As we already used this lowest feasible grid-spacing for simulating such a case (simulating this fog event with Δ=1m occupies 3072 processor units for approximately 48h on a supercomputer).

However, since spurious supersaturations also occur for small grid spacing's since it is more a question of the ratio of advection and condensational phase relaxation time scales we decided to implement and test this method in our model and include the results within this manuscript.

**Modification(chapter 2.2.2 and chapter 4.4):** [..] As this referee comment involve major modifications we kindly refer to the revised manuscript and the attached manuscript, which highlights all changes in comparison to the first version individually.

**RC2(10):** Tab 1 and Part 4 : please add and analyze a new test N0 with Twomey or Cohard and saturation adjustment.

**Author's answer:** We added and analyzed a case with saturation adjustment and the activation scheme of Cohard et al., 1998. Moreover, we also added a case using the prognostic approach by using the same activation scheme. This involves a new chapter, describing the feedback of different supersaturation calculation methods on droplet activation similar to Thouron et al, 2012.

**Modification(chapter 2.2.2 and chapter 4.4):** [..] As this referee comment involve major modifications we kindly refer to the revised manuscript and the attached manuscript, which highlights all changes in comparison to the first version individually.

**RC2(11):** Fig 3 : you say « height averaged » and then 2m and 20m. So what?

**Author's answer:** We agree that this description was wrong. It is a horizontal average at different heights.

**Modification(Fig. 3):** Time series of horizontal [..]

**RC2(12):** Fig.4 : do time marks refer to C1 or REF?

**Author's answer:** Due to major modification's of the manuscript this passages is removed.

**Modification(p?, l?):** [..]

**RC2(13):** P11 l 4 : why are the time steps in the plural? Can you also explain shortly why they are so small?

**Author's answer:** The revised version uses the singular. During the time integration the time step is calculated dynamically. For calculating the length of the new time step our model consider the CFL-criterion (Courant et al., 1928) as well as the diffusion-criterion (e.g. Jacobson, 2005, chap 6.4.4.1)

and afterward takes the minimum of both. Both of them led to a decreased time step by decreasing grid spacing and increasing wind speed. In our cases the grid spacing is relatively small with some moderate wind speed. We had to use a case where the wind speed is strong enough to generate turbulence, otherwise our LES were not able to simulate such a case, which then can favorably be done by DNS.
**Modification(p?, l?):** [..] time step [..]

**RC2(14):** P 12 l 17 : it is C1 minus REF, isn't it?
**Author's answer:** Yes, it is. However, due to major modifications part is removed from the revised manuscript.

**RC2(15):** P12 l 21-22 : How are these higher liquid mixing ratios produced?
**Author's answer:** This is explained by smaller evaporation rates in the case of C1. Due to that the case C1 exhibits in higher levels during the lifting phase of the fog slightly larger values for the liquid water mixing ratio, as evaporation is the dominant process.
**Modification(p?, l?):** [..]as evaporation is the dominant process during the dissipation phase.

**RC2(16):** P 12 l 27 : Again why is the time step approximated?
**Author's answer:** Again, the time step is not fixed. Instead it is calculated new at every time step. Therefore, there is no constant value during one simulation, instead if it is set manually. The latter should only be done if one is sure that the aforementioned criterion are not violated by the manual set time step. But I agree that 'approximately' is the wrong term to describe a well known value. Instead I calculated the average time step of a 4m simulation which was 0.58 s.
**Modification(p12, l7):** [..] on average 0.58 [..]

**RC2(17):** P12 l 26-35 : This paragraph is not acceptable as you conclude on a sensitivity of the time step without showing any result.
**Author's answer:** We removed this paragraph from the manuscript. However, this issue is discussed in more detail by answering the second Referee Comment, what we gladly refer to.
**Modification(p12 l 26-35 ):** [..]Removed this section.

**RC2(18):** P13 l 4 : what is the reference to say that liquid water is overestimated ? Why do not you use the observed value?
**Author's answer:** There is no observed value for this fog event. Our assumptions that the value of the saturation adjustment is overestimated is based on theoretically consideration and on literature found information that conditions for applying saturation adjustment are violated here. However, since this is no evidence for an overestimation in comparison to the real value we replaced this phrase by "higher".
**Modification(p13, l4):** [..] higher in the case of saturation adjustment.

**RC2(19):** Fig 7 : $n_c$ is a 3D field. So is it a vertical and horizontal average, or is it for the first vertical level?
**Author's answer:** It is a horizontal and vertical average for the whole fog layer. Corrected in the revised version.
**Modification(Fig. 7):** [..] (as a horizontal and vertical average of the fog layer) [..]

**RC2(20):** P 14 l 21 : as it is the explicit method, why do you take care of maximum supersaturation?

**Author's answer:** We revised this passage as we must admit that it was confusing to speak about maximum supersaturation for the explicit method, which is commonly used for activation parameterization in case of saturation adjustment. Our aim here was to show that we were able to reproduce typical observed values for the supersaturation. However, for that we do not need to refer to the maximum value. Mainly, those observed values are measured at a height of 2m. Accordingly, in the revised manuscript we connect the observed values with the shown values of simulation in 2m.

**Modification(p?, l?):** [..] while in case EXP and PRG average supersaturation of 0.05% in 2 m occur, which corresponds to typical within fog.

**RC2(21):** What is new from Fig. 9 and 10?
**Author's answer:** In Figure 9 and 10 the microphysical tendiencies are discussed in detail. In contrast to Fig. 5 they consider a full two-moment microphysics scheme, i.e. that also the droplet number concentration is altered. Due to that it could exemplary shown what processes and how strong certain processes influence the

**RC2(22):** p 16 : Could you conclude that the radiation impact of $n_c$ is more important than in the sedimentation process ?
**Author's answer:** This is in interesting objection. Since, we focused here on the impact of microphysical parametrization (and the effect of the radiative impact of $n_c$ is considered within the radiation model) we have not done studies yet to quantify the feedback to e.g. radiative cooling. To isolate this processes (since there is a feedback mechanism: radiative cooling produces higher supersaturation $\rightarrow$ leading to more activated droplets $\rightarrow$ leading to an decreased average radius (sine the surplus water vapor is distributed on more droplets) $\rightarrow$ slower sedimentation and $\rightarrow$ causes stronger radiative cooling, since the effective radius is decreased $\rightarrow$ leading to new (maybe stronger) supersaturation) more studies must be conducted to answer this question appropriately. Moreover, for the sedimentation process a similar feedback mechanism is involved. which might be shortly outlined as: if the number of droplets decrease due to sedimentation $\rightarrow$ the water vapor surplus is distributed on less droplets $\rightarrow$ leading to higher average radius $\rightarrow$ lesser optical thickness and $\rightarrow$ stronger sedimentation.
To get an quantitative idea which of those processes is more important determining the life cycle of the fog would include two more simulation in which the number concentration is kept constant on the one hand for the radiation effect and on the other hand for the sedimentation process.
**Modification:** None.

**RC2(23):** Fig 9 : it would be better to put the total tendency in b than in c, as profiles are too intermingled in c.
**Author's answer:** We agreed and modified the figures as we put the total tendency in an own plot.
**Modification(Fig. 9 & 10)** [..] Modified Fig. 9 and Fig. 10.

**RC2(24):** Fig 10 : Deactivation means evaporation?
**Author's answer:** Yes, it does. Due to reasons of consistency it is adapted to equation 2.
**Modification(FIG10):** [..] deactivation $\rightarrow$ evaporation

**Misspelling :**
- p1 l 20 : aerosols
- p2 l 9 : as as
- p12 l 21 : diminishes
- p14 l 18 : is$\rightarrow$ are

**- p 15 l 16 : shows**

All misspellings are corrected in the revised version.

---

## Referee Report (RR1)

**Second review of the paper acp-2018-1139 « Large-eddy simulation of radiation fog with comprehensive two-moment bulk microphysics: Impact of different aerosol activation and condensation parameterizations» from Johannes Schwenkel and Björn Maronga**

**General comments** : Significant improvements have been brought in this second version and authors have made significant efforts to address criticisms. For instance a prognostic approach of supersaturation has been added and has made a substantial contribution, allowing also to correct a bug. But there are still some weaknesses, inaccuracies and confusion, making the paper not suitable for publication in ACP. Therefore I recommend a second revision before publication.

My major concerns are :
- The sensitivity of the supersaturation parametrization is presented in 2 parts without a clear link between them, and the key conclusions are not clear. Indeed, a first part (4.2) refers to 1-moment microphysical scheme (as nc is fixed) and concludes to the negligible sensitivity of the supersaturation parametrization. But this test is not interested as firstly most of LESs use a 2-moment scheme, and secondly a prognostic saturation is only of interest if droplet concentration is prognostic. It would have no sense if a prognostic saturation scheme was associated with a 1-moment scheme. The second part (4.4) refers to 2-moment scheme and concludes to the importance of supersaturation parametrization as LWP is significantly changed. This 2nd test is the most interested. Additionally, these 2 parts are separated by a sensitivity test of activation parametrization (4.3). Therefore the conclusions are confusing and the paper does not appear beautifully built. From my point of view, the best would be to remove the test of supersaturation parametrization with 1-moment scheme. But if the authors want to keep it as I suppose, it is necessary to merge those parts (with 2 subparts : 1-moment and then 2-moment scheme) and to enhance the conclusion with the 2-moment scheme. The main conclusion will be in agreement with Thouron et al. (2012) with a new aspect concerning application to radiative fog. The conclusion must be revisited too, considering this aspect.
- Concerning the supersaturation parametrization again, that would make it clearer if the method called « explicit supersaturation calculation » was replaced by « diagnostic of supersaturation » to be distinguished from the prognostic approach (as the prognostic approach is also explicit). This would require to replace EXP with DIA in all the text and figures. For the prognostic supersaturation, it is not clear if the supersaturation is advected ? If not, it would be better to call it « pseudo-prognostic » as in Thouron et al. (2012). P9, there is a confusion between $\delta$ and s used previously. What is their difference?
- Concerning the comparison of different activation parametrizations, I remain convinced that it is mainly reduced to a sensitivity test to the CCN concentration as the activation spectra of Fig.A1 show. As authors do not want to change this test for users' need, it is important to insist more on the CCN concentration change. Users must be warned that the choice of the activation method changes significantly the CCN concentration.
- The conclusion needs to be revisited by replacing experiment names with physical terms, and by considering the sensitivity study to saturation scheme mainly for 2-moment schemes, which constitutes the main new result.
- Also there are a lot of mispelling errors. A careful reading by a native english speaker remains necessary.

**More specifically** :

1. p 2 l l6 : you can add a reference to the Meso-NH model : Lac et al., 2018 : Lac, C., J.-P. Chaboureau, et al., Overview of the Meso-NH model version 5.4 and its applications, *Geosci. Model Dev.*, *11*, 1929-1969, 2018.
2. p 2 l 14 : « focusing on the influence of drag effect  **and** droplet deposition »

3. p 2 l18 : Most of the 2-moment schemes used for fog consider radiative cooling as a term of the supersaturation equation. This remark is not relevant.
4. P 2 l 19 « in its **development and** mature stage »
5. P 2 l 28 : add Thouron et al. (2012) to Lebo et al. (2012). Therefore in the next sentence, you can shorten with : « Following these studies ... »
6. p 10 l 5 : do you use cyclic conditions ?
7. P 11 l1-4 : not clear. Please rephrase
8. For parts 4.2, 4.3 and 4.4 the numbering is not correct as you have only a single subpart : 4.2.1, 4.3.1, 4.4.1
9. P 12 l 5 : « In this section … » : necessary to add « with a 1-moment scheme in a LES »
10. P 12 l 16 : instead of Mazoyer et al. (2017) you can add the new reference : Mazoyer et al. (2019) just accepted which is an experimental study : https://www.atmos-chem-phys-discuss.net/acp-2018-875/
11. P 13 l 4 : « drops rapidly **in PRG and EXP** »
12. P 13 l 9 : I do not understand why differences of RH at 2m between SAT and (PRG,EXP) do not lead to differences on dissipation time at the ground.
13. Figure 5 is not nice and subfigures on the right are too small. Is it necessary to present the 3 hours for the right part with the budget ? Only 6 UTC would be sufficient as in Fig. 9.
14. P 14 l 7 : « mature phase **before sunrise**, and mature phase after sunrise »
15. P 14 l 9 : you cannot say that the differences between the runs are negligible as budgets discriminate 2 sets : SAT and (EXP,PRG)
16. p 15 l 13 : « differences for activation **in a 2-moment scheme** might be crucial » : that is why explanations are confusing and parts 4.2 and 4.4 must be merged.
17. P 16 l 15 : this result is not new. As a minimum add a reference as Boutle et al. (2018)
18. p 17 l 1 : where are the observed values ?
19. P 18 l 2 : « N2EXP suffers the most ... » : it is a negative assessment, but what is the reference ?
20. P 18 l 12 : where do you show temporal evolution of supersaturation ?
21. Part 4.4.1 : Is it the same time step for the coarser resolutions ? Otherwise the differences could be due to the impact of the time step instead of the impact of the resolution. If it is the same time step, it is necessary to specify it. If not, you have to run the coarser grids with the same time step (which will not cause instability problems).
22. P 20 l 19, P22 l 2 and P 23 l 1 : « microphysical parametrizations » is too vague and must be replaced by « supersaturation calculation »
23. P 21 : What's about the ratio between N2SAT and N2PRG according to the resolution ?
24. In the conclusion, you have to forget abbreviations N1EXP, N2EXP … and to explain the results in physical terms. Also when you discuss supersaturation calculation, you have to be clear between 1-moment and 2-moment microphysical scheme.
25. P 22 l 22 : add « in agreement with previous studies »

**Misspelling** : there are a lot of errors, the reading was not assiduous. Only a few ones are reported below.

- after a «:», you have to use a lowercase letter : in many parts of the text
- p1 l 2 : cycle
- p 3 l 8 : provide**s**
- p 3 l 18 : start**sed**
- p 12 l 6 : « **which** differs »
- p 20 l 20 : resolution**s**, remove one « the », « comparison with » ...

---

## Author Response (AR2)

Second review of the paper acp-2018-1139
**« Large-eddy simulation of radiation fog with**
**comprehensive two-moment bulk microphysics: Impact of different aerosol activation and**
**condensation parameterizations»**

from Johannes Schwenkel and Björn Maronga

**RC2:** General comments : Significant improvements have been brought in this second version and authors have made significant efforts to address criticisms. For instance a prognostic approach of supersaturation has been added and has made a substantial contribution, allowing also to correct a bug. But there are still some weaknesses, inaccuracies and confusion, making the paper not suitable for publication in ACP. Therefore I recommend a second revision before publication.
**Author's answer:** First of all we would like to thank the reviewer again for the constructive and helpful comments. With this review we have shifted our main focus (what has been missed during the last revision) of this paper to the new introduced results concerning a comparison of different supersaturation calculations in a two-moment microphysics scheme. With the help of these comments, it was possible to contribute to a significant improvement in the work and to clarify paper.

**My major concerns are:**
**RC2:** - The sensitivity of the supersaturation parametrization is presented in 2 parts without a clear link between them, and the key conclusions are not clear. Indeed, a first part (4.2) refers to 1-moment microphysical scheme (as nc is fixed) and concludes to the negligible sensitivity of the supersaturation parametrization. But this test is not interested as firstly most of LESs use a 2-moment scheme, and secondly a prognostic saturation is only of interest if droplet concentration is prognostic. It would have no sense if a prognostic saturation scheme was associated with a 1-moment scheme. The second part (4.4) refers to 2-moment scheme and concludes to the importance of supersaturation parametrization as LWP is significantly changed. This 2nd test is the most interested. Additionally, these 2 parts are separated by a sensitivity test of activation parametrization
(4.3). Therefore the conclusions are confusing and the paper does not appear beautifully built. From my point of view, the best would be to remove the test of supersaturation parametrization with 1- moment scheme. But if the authors want to keep it as I suppose, it is necessary to merge those parts (with 2 subparts : 1-moment and then 2-moment scheme) and to enhance the conclusion with the 2- moment scheme. The main conclusion will be in agreement with Thouron et al. (2012) with a new aspect concerning application to radiative fog. The conclusion must be revisited too, considering this aspect.
**Author's answer:** We agree with that objection. We followed your advice and have restructured the paper as suggested. Moreover, we also clarified our conclusion and set the focus to the two-moment microphysics using different methods for calculating the supersaturation. However, we would like to keep the part with the one-moment microphysics scheme as it was an open question that needed to be addressed. However, we shortened this chapter and put the focus on the main result that the error can be neglected.
**Modification:** As this comment result in a restructuring and major modification of chapter 4 and 5 we would like to refer to the marked-up manuscript in which all modifications are highlighted.

**RC2:** - Concerning the supersaturation parametrization again, that would make it clearer if the method called « explicit supersaturation calculation » was replaced by « diagnostic of supersaturation » to  be distinguished from the prognostic approach (as the prognostic approach is also explicit). This would require to replace EXP with DIA in all the text and figures. For the prognostic supersaturation, it is not clear if the supersaturation is advected ? If not, it would be better to call it « pseudo-prognostic » as in Thouron et al. (2012). P9, there is a confusion between d

and s used previously. What is their difference?

**Author's answer:** We agree with the first point, and have renamed the method, which was previously named by *'explicit supersaturation'* to *'diagnostic supersaturation'*. Therefore, all corresponding figures and text parts were adopted. The objection is correct and hence this modification may improve the comparability with other studies. Concerning the second point: Yes our supersaturation is advected and is therefore represented as an own prognostic quantity. We decided to implement this method (after consultations with a former colleague), which basically follows the implementation described in Morrison and Grabowski, 2008, (but no extra term for inhomogenous mixing). Due to that we would prefer to leave the abbreviation *'PRG'* for the prognostic calculation.

The usage of the absolute supersaturation ($d = q_v - q_s$) is necessary as it simplifies the advection of supersaturation (see Morrison and Grabowski , 2008, section 2b). However, as they mentioned it is also possible to derive a solution for s, but this would involves additional terms and is more complex.

**Modification(p9):** Note, that here the absolute supersaturation is taken, as using s would involve more terms and is more complex to solve (Morrison and Grabowski, 2007).

**RC2:** - Concerning the comparison of different activation parametrizations, I remain convinced that it is mainly reduced to a sensitivity test to the CCN concentration as the activation spectra of Fig.A1 show. As authors do not want to change this test for users' need, it is important to insist more on the CCN concentration change. Users must be warned that the choice of the activation method changes significantly the CCN concentration.

**Author's answer:** We agree with your objection as the choice of activation parametrization significantly change the CCN concentration. Therefore, we emphasized in the revised manuscript that the differences between the activation schemes are produced by different CCN concentrations and this part of the study can be understood as a sensitivity study of different CCN concentration. However, in our opinion, this is a relevant information from the user's perspective should be aware of the fact that different activation parameterizations lead to significantly different CCN concentrations and thus different fog structure.

**Modification:** [..] Using those different parameterizations resulting in different activation spectra, which are shown in Fig.2. One can see, that especially the CCN concentration is changed by using these different methods, such that this part of the study is equivalent to sensitivity study of different CCN concentration but realized by using different coexisting parameterizations. [..]

[..] This part of the study can be regarded as a sensitivity study of different CCN concentrations realized by applying different activation schemes, which is illustrated also in Fig 2. However, from a model user's perspective, such a sensitivity is of great importance as CCN concentrations are usually difficult (case studies) or even impossible (forecasting) to obtain and model results thus might highly depend on the chosen activation parameterization. [..]

[..] However, it must be mentioned that these differences are attributed to the fact that the CCN concentration is different for the investigated schemes. This part of the study can thus also be understood as a sensitivity study for different CCN concentrations realized by the usage of different activation schemes. [..]

**RC2:** - The conclusion needs to be revisited by replacing experiment names with physical terms, and by considering the sensitivity study to saturation scheme mainly for 2-moment schemes, which constitutes the main new result.

**Author's answer:** In the revised manuscript the conclusions have been rewritten and focusing now more on the main result of the two-moment microphysics scheme applying different supersaturation calculations.

**Modification:** As this comment involve major modifications we would kindly refer to the revised

manuscript and to the marked-up manuscript in which all modifications are highlighted.

**RC2:** - Also there are a lot of misspelling errors. A careful reading by a native english speaker remains necessary.
**Author's answer: We are really sorry for having been too sloppy in the writing process. We tried our best to get rid of all language issues.**

**More specifically :**
**RC2:** 1. p 2 l l6 : you can add a reference to the Meso-NH model : Lac et al., 2018 : Lac, C., J.-P. Chaboureau, et al., Overview of the Meso-NH model version 5.4 and its applications, Geosci. Model Dev., 11, 1929-1969, 2018.
**Author's answer:** Done.
**Modification(p2, l16):** Citation added.

**RC2:** 2. p 2 l 14 : « focusing on the influence of drag effect on and droplet deposition »
**Modification(p2, l14):[..]** but using the three-dimensionsl (3D) Large-Eddy Simulation (LES) mode, and focusing on the drag effect of vegetation on droplet deposition.

**RC2:** 3. p 2 l18 : Most of the 2-moment schemes used for fog consider radiative cooling as a term of the supersaturation equation. This remark is not relevant.
**Author's answer:** We agree in a sense that models explicitly used for fog should and will consider radiative cooling as a term of the supersaturation calculation. However, based on the publication of Boutle et al., 2018 some NWP models using or might use activation parameterizations and saturation adjustment without considering radiative cooling explicitly. Instead they assume a minimum vertical velocity updraft and therefore fail for correct number concentration for fog. Hence with this remark we wanted to state that some typical activation schemes are not appropriate (without extensions) for fog.
**Modification(p2, l18):** None.

**RC2:** 4. P 2 l 19 « in its development and mature stage »
**Author's answer:** Agreed.
**Modification(p2, l19):** Rewritten.

**RC2:**5. P 2 l 28 : add Thouron et al. (2012) to Lebo et al. (2012). Therefore in the next sentence, you can shorten with : « Following these studies ... »
**Author's answer:** Agreed.
**Modification(p2, l28):** Rewritten.

**RC2:**6. p 10 l 5 : do you use cyclic conditions ?
**Author's answer:** Yes we do use cyclic conditions at the lateral boundaries.
**Modification(p10, l6):** Cyclic conditions were applied at the lateral boundaries.

**RC2:**7. P 11 l1-4 : not clear. Please rephrase
**Author's answer:** We agree, that the main message of the sentence remains unclear. At this point it should be noted that we have not conducted any further studies in which we have varied the aerosol parameters. However, this note is confusing as it was never claimed that this was the aim of the study and it was given only for parametrization. Therefore, we have decided to delete this sentence completely.
**Modification(p11, l1-4):** Deleted.

**RC2:**8. For parts 4.2, 4.3 and 4.4 the numbering is not correct as you have only a single subpart : 4.2.1, 4.3.1, 4.4.1

**Author's answer:** Thank you for that objection. As a consequence of the restructuring of the result chapter this mistake is fixed. Furthermore, the part concerning the analysis of the budgets is ordered in an own subsection. Due to that chapters with only one single subpart are avoided.
**Modification:** Corrected and changed numbering.

**RC2:**9. P 12 l 5 : « In this section ... » : necessary to add « with a 1-moment scheme in a LES »
**Author's answer:** Agreed and added to the revised manuscript.
**Modification(p12, l5):** In this section we discuss the error introduced by using saturation adjustment for simulating radiation fog with a one-moment scheme in a LES.[...]

**RC2:**10. P 12 l 16 : instead of Mazoyer et al. (2017) you can add the new reference : Mazoyer et al. (2019) just accepted which is an experimental study : https://www.atmos-chem-phys-discuss.net/acp-2018-875/
**Author's answer:** Thank you for that suggestion. We replaced the citation with the currently accepted manuscript by Mazoyer et al.,2018. However, we used the citation as given on the web-page of Atmospheric Chemistry and Physics.
**Modification(p12, l16):** Mazoyer et al. (2017) → Mazoyer et al. (2019)

**RC2:**11. P 13 l 4 : « drops rapidly in PRG and EXP »
**Author's answer:** Agreed and added to the revised manuscript.
**Modification(p13, l4):** Around 0600 UTC, which is shortly after sunrise, relative humidity drops rapidly in PRG and DIA as a direct consequence of direct solar heating of the surface and the near-surface air, preventing further supersaturation at these heights

**RC2:**12. P 13 l 9 : I do not understand why differences of RH at 2m between SAT and (PRG,EXP) do not lead to differences on dissipation time at the ground.
**Author's answer:** This can be attributed to the different methods calculating and allowing supersaturation and liquid water in subsaturated areas. Using the diagnostic and prognostic method, liquid water can also be present in regions where the air is slightly subsaturated. In contrast, using saturation adjustment means the relative humidity is kept to saturation as long as no liquid water is present anymore. A closer look to Figure 3 shows that the times when all schemes reaches a lower relative humidity (Rh<99.8) occurs for all cases simultaneously.
**Modification(p13, l9):** None.

**RC2:**13. Figure 5 is not nice and subfigures on the right are too small. Is it necessary to present the 3 hours for the right part with the budget ? Only 6 UTC would be sufficient as in Fig. 9.
**Author's answer:** As the main focus of our paper has shifted, we decided to remove this figure completely as it does not provide substantial information supporting the main objective of this paper anymore.
**Modification(Figure 5):** Removed.

**RC2:**14. P 14 l 7 : « mature phase before sunrise, and mature phase after sunrise »
**Author's answer:** We agree to this. However, this part was removed from the revised manuscript.
**Modification(p14, l7):** None.

**RC2:**15. P 14 l 9 : you cannot say that the differences between the runs are negligible as budgets discriminate 2 sets : SAT and (EXP,PRG)
**Author's answer:** Your objection is correct. However, for improving the clarity and pushing the focus more on the results regarding different supersaturation calculations in a 2-moment microphysics we decided to remove this part from the manuscript as it is not an essential result.
**Modification(p14,l9):** Deleted.

**RC2:**16. p 15 l 13 : « differences for activation in a 2-moment scheme might be crucial » : that is why explanations are confusing and parts 4.2 and 4.4 must be merged.
**Author's answer:** Agree, and changed accordingly.
**Modification(p15,l13):** Merged.

**RC2:**17. P 16 l 15 : this result is not new. As a minimum add a reference as Boutle et al. (2018) .
**Author's answer:** That is correct. We have added the missing reference.
**Modification(p16,l15):** A linear relationship between LWP and $n_c$ can be found: a higher $n_c$ leads to higher LWP, which is in agreement to other studies as e.g. Boutle et al. (2018).

**RC2:**18. p 17 l 1 : where are the observed values ?
**Author's answer:** They are added in the revised manuscript.
**Modification(Fig 11):** In Fig.11 the simulated visibility for the cases N1DIA-N3DIA  in 2m height together with the observed values at Cabauw (for illustration only).

**RC2:**19. P 18 l 2 : « N2EXP suffers the most ... » : it is a negative assessment, but what is the reference?
**Author's answer:** Definitively "suffers" was the wrong term to describe the finding. Since there is no correct or false within this relative comparison, we decided to refrain from such  assessments. Therefore, the sentence was rephrased.
**Modification(p18, l2):** Therefore, case N2EXP experiences the strongest loss of liquid water due to sedimentation (in relative terms).

**RC2:**20. P 18 l 12 : where do you show temporal evolution of supersaturation?
**Author's answer:** The evolution of the supersaturation is not explicitly shown in this chapter. But it is similar to the one showed in Fig. 3. However, we attached profiles of the supersaturation of 0400 UTC, 0600UTC and 0800UTC to prove our statement. Nevertheless, we would prefer not to include them within the paper, trying to avoid more figures.
**Modification(p18,l12):** None.

**RC2:**21. Part 4.4.1 : Is it the same time step for the coarser resolutions ? Otherwise the differences could be due to the impact of the time step instead of the impact of the resolution. If it is the same time step, it is necessary to specify it. If not, you have to run the coarser grids with the same time step (which will not cause instability problems).
**Author's answer:** Yes, we adopted the time step for the coarser simulations similar to the high resolved simulation, i.e. that simulations with 2m and 4m have a prescribed time step of 0.125s which is the minimum in the high resolved cases. Therefore, we have adopted the manuscript accordingly.
**Modification(Part 4.4.1):** For isolating the effect of the grid spacing, all simulations with a coarser grid spacing were carried out with the same time step of 0.125s, which corresponds to the average time step of the simulations at highest grid spacing of 1m. In this way, effects of different time steps induced by different grid spacings, could be eliminated.

**RC2:**22. P 20 l 19, P22 l 2 and P 23 l 1 : « microphysical parametrizations » is too vague and must be replaced by « supersaturation calculation »
**Author's answer:** We agree on this objection and have modified the revised manuscript accordingly.
**Modification(p20,l19, p22,l2 and p23 l1):** Rephrased.

**RC2:**23. P 21 : What's about the ratio between N2SAT and N2PRG according to the resolution?
**Author's answer:**  We examined the differences, even though as it was not that straight forward as for the N2PRG/N2DIA as the fog life cycle differs stronger. We found that also the relative change

of N2SAT is somewhat larger by decreasing spatial resolution as the relative changes of N2PRG. More precisely, the relative differences (N2SAT/N2PRG), which are quite large anyhow, increases from 81.1% (for the 1 m case), 83.6% by using a grid spacing of 2m up to 93.2%  for the case using a grid spacing of 4m in the mature phase. However, from this analysis it is difficult to say what are the reasons for that, as the fog layer is much more deeper and therefore develop differently.
**Modification(p21):**  None.

**RC2:**24. In the conclusion, you have to forget abbreviations N1EXP, N2EXP ... and to explain the results in physical terms. Also when you discuss supersaturation calculation, you have to be clear between 1-moment and 2-moment microphysical scheme.
**Author's answer:** Thank you for that remark. We have removed all abbreviations from the conclusions. Moreover, we reviewed the whole manuscript and clarifying the conducted studies and explicitly labeled them concerning using one-moment or two-moment microphysics.
**Modification(conclusion):** Rewritten.

**RC2:**25. P 22 l 22 : add « in agreement with previous studies »
Misspelling : there are a lot of errors, the reading was not assiduous. Only a few ones are reported below.
- after a «:», you have to use a lowercase letter : in many parts of the text
- p1 l 2 : cycle
- p 3 l 8 : provides
- p 3 l 18 : startsed
- p 12 l 6 : « which differs »
- p 20 l 20 : resolutions, remove one « the », « comparison with » …
**Author's answer:** All reported misspelling are corrected in the revised revision. Furthermore, we have again double checked the spelling carefully.

**Comment to further improvements. While rechecking the manuscript carefully, we made some additional improvements which are listed below:**

**-** The was a double use of the variables $k$ and $A$. $k$ as a parameter for sedimentation is renamed to $k_F$. $A$ as a parameter for the Activation scheme of Khvorostyanov and Curry (2006)  is renamed by $A_K$.
- The appendix was removed as it was only one figure. Due to that the figure was placed within the manuscript which further helped to clarify the main difference between activation schemes.
- We have rephrased some sentences, as they were grammatical incorrect.
- We unified the axis label of all figure and correct them e.g. Liquid water $\rightarrow$ Liquid water mixing ratio
- The number concentration in case N2SAT in comparison with N2PRG/N2DIA was 60% higher at 0600 UTC. The value of 50% refers to the difference at 0800 UTC. This mistake is corrected in the revised revision.

**Note all changes concerning the last revision are highlighted in a separate file.**

[revised manuscript text omitted]
_{\mathrm{s}}(T_{\mathrm{l}}) = \frac{R_{\mathrm{d}}}{R_{\mathrm{v}}}\frac{e_{\mathrm{s}}(T_{\mathrm{l}})}{p - e_{\mathrm{s}}(T_{\mathrm{l}})}, \tag{12}$$

where $T_\mathrm{l}$ is the liquid water temperature and $p$ is pressure.  $R_\mathrm{d}$ and $R_\mathrm{v}$ are the specific gas constants for dry air and water vapor , respectively. For the saturation vapor pressure $(e_\mathrm{s})$ an empirical relationship of Bougeault (1981) is used. In a second step, $q_\mathrm{s}$ is corrected using a first-order Taylor series expansion of $q_\mathrm{s}$:

$$q_\mathrm{s}(T) = q_\mathrm{s}(T_\mathrm{l}) \frac{1 + \beta q}{1 + \beta q_\mathrm{s}(T_\mathrm{l})} \frac{1 + \gamma q}{1 + \gamma q_\mathrm{s}(T_\mathrm{l})}, \tag{13}$$

with

$$\beta\gamma = \frac{L_\mathrm{v}}{R_\mathrm{v} c_\mathrm{p} T_\mathrm{l}^2}, \tag{14}$$

[revised manuscript text omitted]